# DesignX: Human-Competitive Algorithm Designer for Black-Box Optimization

**Hongshu Guo**[1], **Zeyuan Ma**[1], **Yining Ma**[2],
**Xinglin Zhang**[1], **Wei-Neng Chen**[1], **Yue-Jiao Gong**[1,*]
[1]South China University of Technology
[2]Massachusetts Institute of Technology
{guohongshu369, scut.crazynicolas}@gmail.com, yiningma@mit.edu
{csxlzhang, cschenwn}@scut.edu.cn, gongyuejiao@gmail.com

## Abstract

Designing effective black-box optimizers is hampered by limited problem-specific knowledge and manual control that spans months for almost every detail. In this paper, we present *DesignX*, the first automated algorithm design framework that generates an effective optimizer specific to a given black-box optimization problem within seconds. Rooted in the first principles, we identify two key sub-tasks: 1) algorithm structure generation and 2) hyperparameter control. To enable systematic construction, a comprehensive modular algorithmic space is first built, embracing hundreds of algorithm components collected from decades of research. We then introduce a dual-agent reinforcement learning system that collaborates on structural and parametric design through a novel cooperative training objective, enabling large-scale meta-training across 10k diverse instances. Remarkably, through days of autonomous learning, the DesignX-generated optimizers continuously surpass human-crafted optimizers by orders of magnitude, either on synthetic testbed or on realistic optimization scenarios such as Protein-docking, AutoML and UAV path planning. Further in-depth analysis reveals DesignX's capability to discover non-trivial algorithm patterns beyond expert intuition, which, conversely, provides valuable design insights for the optimization community. We provide DesignX's Python project at https://github.com/MetaEvo/DesignX.

## 1 Introduction

Black-box optimization (BBO) lies at the core of scientific and industrial advances, such as electronic design automation [1], molecular design [2] and AutoML [3]. Yet, BBO is challenging due to unavailable objectives and derivatives, and complex, diverse properties that demand extensive expert knowledge. Evolutionary Computation (EC) is widely recognized as a robust derivative-free paradigm for BBO [4]. Since the 1990s, numerous EC variants such as genetic algorithms[5], differential evolution [6], particle swarm optimization [7], and evolution strategies [8] have emerged. Despite shared core paradigm, they rely on expert-designed adaptive operators [9] and hyperparameter control [10] to achieve the best performance on a particular BBO class or instance.

However, manually redesigning optimizers for each new BBO problem is neither scalable nor practical. Recently, an emerging research avenue termed as Meta-Black-Box-Optimization (MetaBBO) [11] has emerged, which automates algorithm design (AAD) through a bi-level paradigm: a meta-level learns a policy to guide low-level BBO optimizer. By meta-training [12] over a distribution of problems, MetaBBO can generate customized algorithms for both seen and unseen instances.

---

[*]Yue-Jiao Gong is the corresponding author.

39th Conference on Neural Information Processing Systems (NeurIPS 2025).

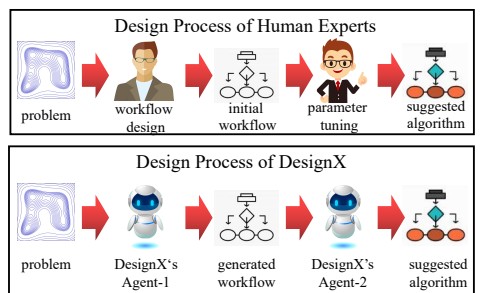 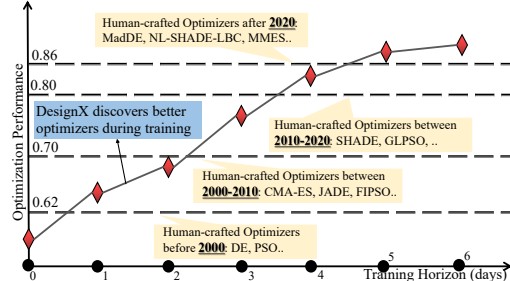

Figure 1: **Left**: Compared to manual design process, DesignX replaces human experts by two learnable agents. **Right**: Four dashed lines denote average performances of well-known human-crafted optimizers in decades. During pre-training, DesignX surprisingly discovers powerful optimizers superior to the ones crafted by human experts.

Despite the success, existing MetaBBO approaches merely focus on learning specific sub-tasks of AAD for EC. Specifically, optimizer design involves two sequential sub-tasks (see Figure 1, top left): (1) determining the algorithm workflow, and (2) (dynamic) algorithm configuration. Existing work addresses the former via algorithm selectors [13–15] or workflow generators [16–19], and the latter through reinforcement learning (RL) [20] for online control [21–24]. While learning a single sub-task eases training, it often results in sub-optimal designs and limits potential performance gains.

In this paper, we advance MetaBBO research by proposing the first unified framework that jointly learns both sub-tasks of algorithm design: workflow generation and (dynamic) algorithm configuration, so as to enable the discovery of human-competitive optimizers in an end-to-end fashion.

This is achieved through several key innovations. Firstly, we extend and enrich the Modular-BBO modularization system in [24], resulting in a more comprehensive system: Modular-EC. Specifically, since Modular-BBO is primarily constructed for Differential Evolution (DE) optimizer, Modular-EC integrates more diverse sub-modules in Evolutionary Strategy (ES), Genetic Algorithm (GA) and Particle Swarm Optimization (PSO) algorithms into its sub-module library. Modular-EC now supports representing different optimizer types, enhancing the capacity of Modular-BBO. Building on the upgraded Modular-EC, we develop a dual-agent reinforcement learning system (see Figure 1, bottom left), where both agents are Transformer-based [25]: 1) Agent-1 autoregressively samples valid optimizer workflows conditioned on the problem instance; 2) Agent-2 dynamically adjusts hyperparameters during optimization by incorporating real-time feedback. A novel cooperative reward scheme encourages both agents to make mutually conditioned decisions, jointly optimizing for maximum performance. We train this dual-agent system on a large-scale problem set of 10k synthetic instances, and observe it consistently discovering optimizers that outperform expert-crafted baselines (see Figure 1, right). Remarkably, through days of autonomous learning, the DesignX-generated optimizers continuously surpass human-crafted optimizers by orders of magnitude, either on synthetic testbed or on realistic optimization scenarios such as Protein-docking, AutoML and UAV. Furthermore, the testing results clearly demonstrate the novelty and superiority of DesignX against up-to-date MetaBBO baselines. To summarize, the contributions of this paper are in three folds:

- To the best of our knowledge, this paper the first deep learning-based framework that jointly learns workflow generation and dynamic algorithm configuration in an end-to-end fashion.
- We obtain a well-performing model (DesignX) through large-scale training, capable of designing powerful optimizers for diverse, unseen, realistic problems.
- Further in-depth analysis reveals the importance of the proposed novel designs, providing first-hand insights on non-trivial algorithm patterns beyond expert intuition.

## 2 Related Works

We review the development of Automated Algorithm Design (AAD) over the past decades. Early efforts by Schmidhuber et al. [26] applied Genetic Programming (GP) to recursively improve another GP in a self-referential manner. Later, GP was applied to design full algorithm templates [27], but difficulties in genotype design and expensive evaluations limited its scalability for BBO problems. Recent MetaBBO approaches integrate machine learning techniques such as reinforcement learning

(RL) and large language models (LLMs) to develop more flexible and generalizable optimizers [11, 28]. These RL-based methods like DEDQN [14] and DEDDQN [13] focused on operator selection and hyperparameter control within fixed algorithm structures. More recent methods leverage Transformer architectures for enhanced control [23, 29], including ConfigX [24] and Q-Mamba [30], which implement online and offline RL, respectively. Other works explored Transformer-based generation of algorithm components. SYMBOL [31] learned to compose new operators as symbolic sequences. ALDes [17] tokenized common algorithmic modules and turned workflow design into sequence generation. GLHF [32] simulated DE operators with trainable modules optimized through gradient descent. Besides these works, the lifelong learning ability of MetaBBO and its adaptability for expensive evaluation condition are also discussed in recent literature [33, 34], as well as the automatic feature learning [35] and training distribution construction [36]. Though these MetaBBO approaches use relatively small neural networks for meta-learning, they achieved tailored performance on specific algorithm design tasks. With LLM-scale models, capabilities expand further. LLMs can search reward functions [37], optimize neural architectures [38], act as optimizers based on previous search trajectories [39], or generate algorithm code from problem descriptions [18, 19, 40]. However, existing work focuses on only one sub-task of AAD: either generating workflows or controlling parameters. No prior method jointly addresses both, which motivates our proposed DesignX to enable end-to-end algorithm design.

## 3 Methodology

### 3.1 Modular-EC

Existing EC optimizers commonly comprise a series of algorithm modules. A massive array of novel algorithm modules have been proposed in literature for specific optimization scenarios [9, 41, 42]. It is a quite natural idea to "stand on the shoulder of giants" for designing new optimizers, that is to say, construct a modular algorithmic space and search for well-performing optimizer workflow in it [43, 44]. Following such idea, ConfigX [24] proposes a comprehensive modularization system: Modular-BBO for learning universal hyper-parameter control policy in DE. It groups commonly used sub-module variants in existing DE optimizers into 9 module types: 6 of which are UNCONTROLLABLE without hyper-parameters: INITIALIZATION [45], BOUNDARY_CONTROL [46], SELECTION [47], NICHING [48], RESTART_STRATEGY [49], POPULATION_REDUCTION [50], and the rest 3 of which are CONTROLLABLE with hyper-parameters: MUTATION [51], CROSSOVER [52], and INFORMATION_SHARING [53].

In Modular-EC, we have added a novel module type OTHER_UPDATE [54, 55] into Modular-BBO's module library, which belongs to CONTROLLABLE namespace. We integrate popular reproduction operators of diverse ES, GA and PSO optimizers into OTHER_UPDATE and also update the other 9 module types by adding corresponding sub-modules in ES, GA and PSO. To summarize, Modular-EC supports 10 module types with 116 module variants in total. This results in millions of possible algorithm workflows, significantly enhancing the expressiveness of Modular-BBO.

For a concrete module variant, Modular-EC assigns it an unique 16-bit binary code *id* for identify. A *topology_rule* list is built within each module variant to indicate which module types are allowed to be placed right after this module variant, ensuring legal generation of optimizer workflow in auto-regressive fashion. We list some examples here: 1) Any EC optimizer must start with INITIAL-IZATION; 2) BOUNDARY_CONTROL is not allowed placed between two subsequent reproduction modules (e.g., MUTATION and CROSSOVER); 3) RESTART_STRATEGY is only allowed to be placed at the end of a EC optimizer. We provide more details of the hierarchical architecture, module variants information of Modular-EC in Appendix A.

### 3.2 Dual-agent Algorithm Design System

We propose a dual-agent algorithm design system for DesignX to operate on Modular-EC. As shown in Figure 2, the system consists of two Transformer-based RL agents: Agent-1 ($\pi_\phi$) and Agent-2 ($\pi_\theta$), each addressing a core sub-task in automated algorithm design. **1) Algorithm workflow generation:** Agent-1 constructs a customized optimizer workflow based on the given problem. **2) Hyperparameter control:** Agent-2 dynamically adjusts the hyperparameters during the optimization process to enhance performance. By jointly addressing both sub-tasks, DesignX offers a more complete and effective solution than methods focusing on only one aspect (see Section 2).

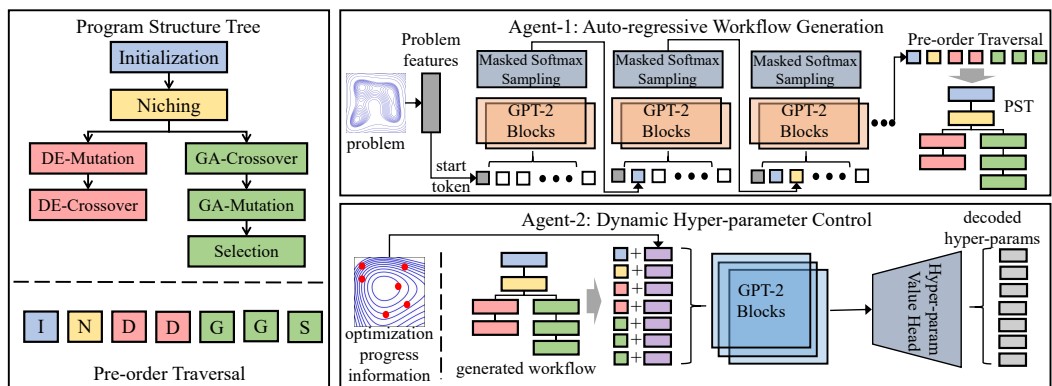

Figure 2: **Left**: The dual-agent system in DesignX processes an optimizer workflow by the pre-order traversal of its program structure tree. **Top Right**: Agent-1 generates legal optimizer workflow in an auto-regressive fashion. **Bottom Right**: Agent-2 controls hyperparameters of the generated optimizer workflow by conditioning on the optimization progress information.

Before we get into further technical details, we first explain the Program Structure Tree (PST) [56] of an algorithm workflow and pre-order traversal of PST. We illustrate a simple example in the left of Figure 2, where a two-population niching-based EC optimizer is represented by PST and corresponding pre-order traversal respectively. The pre-order traversal representation of an optimizer workflow is primarily used in Agent-1 and Agent-2 to align with information processing logic of Transformer architecture, where each module in the traversal is regarded as a token.

### 3.2.1 Agent-1: Workflow Generation

Agent-1's workflow is shown in the top right of Figure 2. Given the feature vector $\mathcal{F}_p$ of an optimization problem $p$, Agent-1 auto-regressively samples module variants from Modular-EC to construct a complete optimizer workflow $\mathcal{A}p = \pi_\phi(\mathcal{F}p)$. The architecture of $\pi_\phi$ consists of four components: 1) a problem feature embedder $\mathcal{W}_{feature} \in \mathbb{R}^{13 \times h}$, where 13 is $\mathcal{F}_p$'s dimension and $h$ denotes the token embedding dimension; 2) a Tokenizer $\mathcal{W}_{token} \in \mathbb{R}^{16 \times h}$, where 16 denotes the 16 bits module *id*; 3) $L$ sequential GPT-2 [57] blocks with $k$ heads and hidden dimension $h$. We use $MSA_1$ to denote these attention blocks; 4) a masked Softmax module $\mathcal{W}_{sample} \in \mathbb{R}^{h \times 117}$, where 117 is the numbers of tokens (116 modules in Modular-EC and an additional *end* token).

**Problem Feature Embedding.** The raw feature $\mathcal{F}_p$ for a given optimization problem $p$ is a 13-dimensional vector, which is further divided into two parts: 1) 4 basic properties: the dimension, allowed maximum function evaluations, upperbound and lowerbound of searching range; 2) 9 statistical properties: we use a well-known optimization problem statistical analysis framework, Exploratory Landscape Analysis (ELA) [58], which provides many statistical low-level features for profiling high-level optimization properties such as multi-modality, separability, global structure, etc. Specifically, we select 9 ELA features with both significant independence and efficient computation according to the sensitivity analysis of ELA features in [59, 60]. We provide a detailed elaboration on these ELA features in Appendix B.1. Once $\mathcal{F}_p$ is obtained, we use $\mathcal{W}_{feature}$ to map it to a $h$-dimensional token, which we denote as *start* for subsequent optimizer workflow generation.

**Auto-regressive Generation.** Starting from the *start* token for problem $p$, Agent-1 auto-regressively generates the pre-order traversal of an optimizer workflow $\mathcal{A}_p$. Suppose Agent-1 has generated $m$ modules $\{\mathcal{A}_p^1, \mathcal{A}_p^2..., \mathcal{A}_p^m\}$, then the sampling distribution of $(m+1)$-th module $\mathcal{A}_p^{m+1}$ is:

$$
\begin{aligned}
P(\mathcal{A}_p^{m+1}|start, \mathcal{A}_p^1, ..., \mathcal{A}_p^m) &\sim Softmax(mask(\mathcal{A}_p^m) \odot (\mathcal{W}_{sample}^{\mathrm{T}} \cdot H^{(m)})), \\
H &= MSA_1(Pos + \{start, \mathcal{W}_{token}^{\mathrm{T}} \cdot \mathcal{A}_p^1.get\_id(), ..., \mathcal{W}_{token}^{\mathrm{T}} \cdot \mathcal{A}_p^m.get\_id()\})
\end{aligned}
\tag{1}
$$

where we first get each sampled module's *id* and use the tokenizer to map them to tokens with $h$-dimension. Then all tokens including *start* are added with Cosine Position Encoding $Pos$. After going through the GPT-2 blocks $MSA_1$, we use $\mathcal{W}_{sample}$ to map the output embedding $H^{(m)}$ for $m$-th module as the prediction head. Recall that we have to ensure the generated workflow is legal. To achieve this, we propose a masked Softmax sampling procedure. A boolean mask vector $mask(\mathcal{A}_p^m) \in \mathbb{R}^{117}$ is obtained by checking $\mathcal{A}_p^m$'s topology rule $\mathcal{A}_p^m.get\_rule()$. Hadamard product

between the mask and prediction head squeezes the sampling probability of illegal modules to 0. We note that the dimension of prediction head and the mask is 117, which corresponds to the 116 modules in Modular-EC and the *end* token. Without the *end* token, Agent-1 has risks of generating infinite trajectory. Refer to Appendix A, Table 3 to check which modules could be placed right before $end$. In the rest of this paper, we use $\pi_\phi(\mathcal{A}_p)$ to denote the sampling probability of a concrete workflow $\mathcal{A}_p$, which is the successive multiplication of all generation steps:

$$\pi_\phi(\mathcal{A}_p) = P(\mathcal{A}_p^1|start)P(\mathcal{A}_p^2|start, \mathcal{A}_p^1)...P(end|start, \mathcal{A}_p^1, ..., \mathcal{A}_p^M) \tag{2}$$

### 3.2.2 Agent-2: Dynamic Algorithm Configuration

Agent-2's workflow is shown in the bottom right of Figure 2. Once $\mathcal{A}_p$ is generated by Agent-1, it is used to optimize $p$. During the optimization process, given some observed optimization progress information $\mathcal{O}_t$ at $t$-th optimization step, Agent-2 dynamically adjusts hyper-parameter values $\mathcal{C}_t = \pi_\theta(\mathcal{O}_t)$ for all CONTROLLABLE modules in $\mathcal{A}_p$. The motivation behind Agent-2 is that: a common observation in EC domain reveals that hyperparameter values in an optimizer more or less impact the exploration/exploitation tradeoff [9]. An effective parameter control policy could further enhance the optimization performance of the optimizer generated by Agent-1.

To suggest per optimization step hyperparameter values for CONTROLLABLE modules in $\mathcal{A}_p$, an informative optimization progress feature vector $\mathcal{O}_t$ is first computed following the common idea of up-to-date MetaBBO approaches [23, 24, 31]. $\mathcal{O}_t$ is a 9-dimensional vector of which each dimension is a statistical feature indicating the local/global distribution in solution/objective space, convergence progress and optimization budget usage information. We provide detailed description of these features in Appendix B.2. Agent-2 then embeds $\mathcal{O}_t$ into each module in $\mathcal{A}_p$ to get all module's embeddings:

$$Emb(\mathcal{A}_p^m) = Pos + \mathcal{W}_{emb}^T \cdot [\mathcal{A}_p^m.get\_id(), \mathcal{O}_t] \quad m = 1, 2..., M \tag{3}$$

where $\mathcal{W}_{emb}^T \in \mathbb{R}^{25 \times h}$ maps the concat of module *id* and $\mathcal{O}_t$ to $h$-dimensional embeddings. The final embedding for each module is obtained by adding the $h$-dimensional embeddings with Cosine Positional Embedding codes, which inject relative order information, to let Agent-2 grasps the overall optimizer workflow structure. The suggested hyperparameter values at optimization step $t$ is decoded by first feeding the embeddings of all modules into $L$ sequential GPT-2 [57] blocks with $k$ heads and hidden dimension $h$ (denoted as $MSA_2$). Then the output decision embeddings $H_{dec}$ are further decoded into normal distribution parameters:

$$\mu = \mathcal{W}_\mu^T \cdot H_{dec}, \quad \Sigma = \mathcal{W}_\Sigma^T \cdot H_{dec}, \quad H_{dec} = MSA_2(Emb(\mathcal{A}_p^1), ..., Emb(\mathcal{A}_p^M)) \tag{4}$$

where $\mathcal{W}_\mu^T \in \mathbb{R}^{h \times N_{max}}$ and $\mathcal{W}_\mu^T \in \mathbb{R}^{h \times N_{max}}$ are network parameters of the hyperparameter value head. They map $H_{dec}$ to the mean parameters $\mu \in \mathbb{R}^{M \times N_{max}}$ and covariance parameters $\Sigma \in \mathbb{R}^{M \times N_{max}}$, where $\mu^{(m)} \in \mathbb{R}^{N_{max}}$ and $\Sigma^{(m)} \in \mathbb{R}^{N_{max}}$ denotes distribution parameters for $m$-th module in $\mathcal{A}_p$. At last, the hyperparameter values $C_t$ are sampled from the predicted normal distributions for all $M$ modules:

$$C_t = \{C_t^1, ..., C_t^M\} \sim \{\mathcal{N}(\mu^{(1)}, \Sigma^{(1)}), ..., \mathcal{N}(\mu^{(M)}, \Sigma^{(M)})\} \tag{5}$$

We have to note that since different modules in Modular-EC might hold different number of hyperparameter values, we predefine a maximum configuration size $N_{max}$ to cover them. If the number of hyper-parameters in a module is less then $N_{max}$, we use the first few sampled values and ignore the rest. Suppose the optimization horizon for problem $p$ is $T$ steps, Agent-2 will be asked $T$ times for deciding the per-step hyper-parameter values. In the rest of this paper, we use $\pi_\theta(C_t|\mathcal{A}_p)$ to denote the associate probability of the hyperparameters for $\mathcal{A}_p$ at optimization step $t$[2]:

$$\pi_\theta(C_t|\mathcal{A}_p) = \prod_{m=1}^M \mathcal{N}(\mu^{(m)}, \Sigma^{(m)}) \tag{6}$$

### 3.3 Cooperative Large Scale Training

We propose a large scale meta-reinforcement-learning paradigm to ensure the pre-trained DesignX model could benefit from the harmonious cooperation between Agent-1 & 2, and is capable of being generalized towards unseen problems.

---

[2]We only consider sampling for modules with at least one hyperparameter.

**Large Scale Synthetic Problem Set.** We construct a large scale synthetic problem set containing 12800 diverse problem instances for the ease of training generalizable DesignX model. 32 representative basic problems are first collected from popular BBO benchmarks [61, 62], including Rastrigin, Schwefel, Rosenbrock, etc. We follow the steps below to generate 12800 diverse problem instances: 1) We first define three problem construction modes, "single", "composition" and "hybrid". "single" mode randomly selects one basic problem. "composition" mode randomly aggregates 2-5 basic problems by weighted summation of their objective functions. "hybrid" mode divides decision variables into some subcomponents and then randomly selects a group of basic functions, which are used for different subcomponents. 2) By randomly selecting the construction modes and determining the searching range, dimension (5-50d), maximum allowed optimization budget (10000-50000 maxFEs) and rotation/shift in solution space, we construct 12800 problem instances with diverse optimization properties, which aligns with the intricate problem distribution in real world. We further randomly split them into a training problem set $\mathcal{D}_{train}$ (9600 instances) and a testing set $\mathcal{D}_{test}$ (3200 instances). A more detailed elaboration is provided in Appendix C.

**Cooperative Training Objective.** We formulate the automated algorithm design task of DesignX as a dual-agent Markov Decision Process (MDP). For each problem instance $p \in \mathcal{D}_{train}$, Agent-1 first generates a legal optimizer workflow $\mathcal{A}_p$ with probability $\pi_\phi(\mathcal{A}_p)$ in Eq. (2). $\mathcal{A}_p$ is then used to optimize $p$ until its allowed optimization budget is used up. For each optimization step $t$ along this optimization process ($T$ steps in total), Agent-2 continuously dictates hyperparameters $C_t$ with probability $\pi_\theta(C_t|\mathcal{A}_p)$ in Eq. (6). We record the reward obtained at $t$-th step as $r_t = \frac{f_p^{t-1,*} - f_p^{t,*}}{f_p^{0,*} - f_p^*}$, where $f_p^{t,*}$ denotes the optimal objective value found until $t$-th step (w.l.o.g., $p$ is assumed as a minimization problem), $f_p^*$ denotes the optimal objective value of $p$. Then the training objective of DesignX's MDP can be formulated as:

$$\mathcal{J}(\phi,\theta) = \mathbb{E}_{p \sim \mathcal{D}_{train}}\left[\sum_{t=1}^{T} r_t\right] = \frac{1}{|\mathcal{D}_{train}|} \sum_{i=1}^{|\mathcal{D}_{train}|} \sum_{t=1}^{T} r_t \tag{7}$$

which is the expected optimization performance if we use DesignX's Agent-1 & 2 to design optimizers for solving problem instances in $\mathcal{D}_{train}$. For Agent-1, there is no intermediate reward (delayed-reinforcement task), hence we train it by episodic reinforcement learning method REINFORCE [63]. For Agent-2, the per-step reward $r_t$ can be used hence we train it by the popular PPO method [64]. We provide the pseudo code of the training procedure in Appendix D, Alg. 1.

# 4 Experimental Analysis

In this section, we discuss the following research questions: **RQ1**: Can DesignX automatically design human-competitive BBO optimizers that excel at both synthetic and realistic scenarios? **RQ2**: What design skills has DesignX learned? **RQ3**: How do the core components in DesignX contribute? **RQ4**: How is the scalability of DesignX in terms of the scaling law? Below, we first introduce the experimental setup and then address RQ1~RQ4 respectively.

**Experiments Setup**. The baselines in experiments include: **1) a DesignX model** trained after 6 days; **2) up-to-date MetaBBO** approaches GLHF [32], DEDQN [14] and GLEET [23] that excel at workflow learning or hyper-parameter control; **3) representative human-crafted optimizers**: a) those before 2000, GA [5], PSO [7] and DE [6]. b) those in 2000-2010, CMAES [65], FIPSO [66], SaDE [67], CLPSO [68] and JADE [69]. c) those in 2010-2020, CoDE [70], IPSO [71], SHADE [72], LM-CMA-ES [73] and GLPSO [74]. d) those after 2020, MadDE [75], jDE21 [76], MMES [77] and NL-SHADE-LBC [78]. For evaluation fairness, we train DesignX and other MetaBBO baselines on the same $\mathcal{D}_{train}$ (see Section 3.3). We leave detailed training settings and other hyper-parameter settings of all baselines at Appendix E.1 & E.2. To simplify presentation, we use following tags: "MetaBBO", "'before 00', "00s", "10s" and "after 20" to tag these baselines.

## 4.1 Performance Comparison (RQ1)

**In-distribution Generalization.** All baselines are tested on our proposed $\mathcal{D}_{test}$ (see Section 3.3), with 51 independent runs for each problem instance. Due to the space limitation, we present the absolute optimization performance of all baselines on 20 of the 3200 tested instances in Table 1. These 20 instances are randomly selected to showcase their diversity in: a) optimization properties,

Table 1: The in-distribution generalization performance in terms of absolute optimization performance results on $\mathcal{D}_{test}$. The best is labeled in green and the second best is labeled in red.

| | before 00 | 00s | 10s | after 20 | MetaBBO | DesignX |
|---|---|---|---|---|---|---|
| F1 MAH, 50D, 30000 FEs | 6.60E+00 ±3.74E+00+ | 1.64E+00 ±1.64E+00+ | 1.27E+00 ±4.41E-01+ | 5.32E+00 ±3.70E+00+ | 2.80E+00 ±0.00E+00+ | **2.89E-01 ±3.93E-01** |
| F79 UAH, 5D, 50000 FEs | 2.98E+00 ±9.95E-01+ | 3.70E+00 ±1.71E+00+ | 5.38E+00 ±4.05E-01+ | 1.81E+00 ±1.83E-01+ | 9.95E-01 ±0.00E+00+ | **5.68E-02 ±1.17E+00** |
| F125 UAH, 10D, 40000 FEs | 1.39E-03 ±1.38E-03+ | 3.50E-06 ±3.50E-06+ | 1.48E-04 ±1.33E-04+ | 1.69E-05 ±7.99E-06+ | 1.08E-04 ±0.00E+00+ | **4.81E-07 ±2.66E-07** |
| F154 UAH, 50D, 10000 FEs | 1.35E+03 ±2.26E+02+ | 1.44E+03 ±3.45E+02+ | 1.38E+03 ±2.40E+02+ | 1.46E+03 ±6.17E+02+ | **5.47E+02 ±0.00E+00−** | 6.99E+02 ±7.45E+01 |
| F211 MAH, 5D, 40000 FEs | 6.55E-01 ±2.92E-01+ | 8.04E-01 ±6.99E-01+ | 2.64E-01 ±9.96E-02+ | 1.28E-01 ±2.43E-02+ | 1.59E-01 ±0.00E+00+ | **7.28E-02 ±6.56E-02** |
| F240 MWL, 20D, 20000 FEs | 6.39E+00 ±4.25E+00+ | 8.72E+00 ±1.71E+00+ | 8.24E+00 ±2.19E+00+ | 3.97E+00 ±3.31E+00+ | 2.05E+00 ±0.00E+00+ | **1.27E-01 ±2.99E+00** |
| F326 UAL, 10D, 40000 FEs | 1.10E+00 ±1.22E-01+ | 1.15E+00 ±4.53E-01+ | 2.47E+00 ±4.64E-01+ | 7.66E-01 ±5.10E-02+ | 1.22E+00 ±0.00E+00+ | **5.84E-01 ±1.66E+00** |
| F411 UAL, 10D, 50000 FEs | 2.50E-01 ±9.51E-02+ | 4.07E-01 ±9.27E-02+ | 2.68E-01 ±7.41E-02+ | 1.28E-01 ±1.11E-02+ | 1.87E-01 ±0.00E+00+ | **7.91E-02 ±4.83E-02** |
| F545 UWL, 5D, 40000 FEs | 2.98E+00 ±9.94E-01+ | 1.49E+00 ±4.97E-01+ | 3.36E+00 ±6.23E-01+ | 8.41E-01 ±1.54E-01+ | 1.99E+00 ±0.00E+00+ | **2.61E-08 ±6.48E-01** |
| F1045 MWH, 10D, 40000 FEs | 7.53E+02 ±1.68E+00+ | 4.20E+02 ±1.47E+02+ | 2.29E+02 ±6.06E+01+ | 1.71E+02 ±1.41E+01+ | 9.21E+02 ±0.00E+00+ | **1.67E+02 ±1.06E+02** |
| F1139 MAH, 10D, 50000 FEs | 1.16E+01 ±1.00E+01+ | 3.95E+00 ±1.67E+00+ | 8.90E+00 ±1.54E+00+ | 2.67E+00 ±1.15E+00+ | 1.19E+01 ±0.00E+00+ | **1.39E-04 ±1.47E+00** |
| F1200 MAL, 50D, 40000 FEs | 7.47E+00 ±6.29E+00+ | 1.14E+00 ±7.00E-03+ | 1.15E+00 ±2.56E-02+ | 1.33E+01 ±1.22E+01+ | 1.27E+00 ±0.00E+00+ | **1.09E+00 ±2.12E-02** |
| F1556 MAH, 10D, 40000 FEs | 3.99E+02 ±3.75E+02+ | 2.03E+02 ±1.76E+02+ | 2.73E+01 ±7.63E+00+ | 1.48E+01 ±7.13E-01+ | **9.45E+00 ±0.00E+00−** | 1.01E+01 ±1.13E+02 |
| F1653 MAH, 20D, 10000 FEs | 2.55E+01 ±1.53E+00+ | 2.53E+01 ±7.11E-01+ | 2.72E+01 ±7.28E-01+ | 1.85E+01 ±2.56E+00+ | 1.69E+01 ±0.00E+00+ | **1.54E+01 ±3.47E+00** |
| F1687 MAL, 50D, 40000 FEs | 8.98E+00 ±6.60E+00+ | 2.07E+01 ±1.07E+01+ | 4.49E-01 ±2.38E-01+ | 2.94E+01 ±2.81E+01+ | 1.94E+00 ±0.00E+00+ | **2.24E-02 ±9.09E+00** |
| F2068 MWH, 20D, 20000 FEs | 3.79E+01 ±6.53E+00+ | 2.32E+00 ±1.13E-01+ | 1.46E+01 ±1.40E+01+ | 1.65E+01 ±1.41E+01+ | 3.72E+01 ±0.00E+00+ | **5.16E-01 ±1.06E+01** |
| F2390 MAL, 10D, 30000 FEs | 3.93E+00 ±2.15E+00+ | 2.78E+00 ±0.00E+00+ | 6.34E+00 ±9.03E-01+ | 1.54E+00 ±1.10E+00+ | 2.04E+01 ±0.00E+00+ | **1.85E-03 ±2.45E+00** |
| F2473 MAL, 10D, 20000 FEs | 1.10E+00 ±9.06E-02+ | 3.98E-01 ±6.88E-02+ | 8.72E-01 ±1.80E-02+ | 6.69E-01 ±2.53E-01+ | **1.42E-01 ±0.00E+00−** | 1.63E-01 ±1.59E-01 |
| F2895 MWL, 10D, 50000 FEs | 1.90E+01 ±3.88E+00+ | 4.34E+00 ±1.66E+00+ | 1.18E+01 ±5.35E+00+ | 4.23E+00 ±5.26E-01+ | 4.98E+00 ±0.00E+00+ | **1.99E+00 ±3.34E+00** |
| F2986 MAL, 50D, 10000 FEs | 4.37E+02 ±1.71E+02+ | 4.93E+02 ±3.74E+02+ | 1.60E+02 ±6.45E+01+ | 2.51E+03 ±2.42E+03+ | 1.01E+02 ±0.00E+00+ | **8.90E+01 ±2.65E+01** |
| Normalized Averaged Objective | 2.94E-01 ±1.01E+00+ | 1.96E-01 ±1.62E+00+ | 1.54E-01 ±2.61E-01+ | 1.46E-01 ±2.35E-01+ | 1.32E-01 ±7.36E-01+ | **8.26E-02 ±1.75E-01** |

Figure 3: The generalization performance of baselines on realistic scenarios.

"U/M" for unimodal/multi-modal, "A/W" for adequate or weak global structures and "L/H" for low or high conditioning; b) problem dimensions, 5D-50D; c) allowed optimization budget in terms of the number function evaluations (FEs). We additionally average the baselines in each tag ("before 00", "00s" and etc.) for the ease of presentation.

The results in Table 1 reveal that: 1) The human-crafted BBO optimizers achieve progressive advancement through the expert-level designs proposed over the past decades. However, they are sill restricted by *no-free-lunch* theorem. 2) By incorporating learning paradigm into BBO optimizers, MetaBBO approaches are capable of boosting the low-level optimizers on some problem instances. 3) The optimization performance of DesignX surpasses both MetaBBO and hand-crafted BBO baselines, ranking the first place on almost all tested instances with diverse properties. Through learning the bi-agent system across a large scale problem distribution ($\mathcal{D}_{train}$), DesignX intelligently designs powerful and customized optimizers for different problems. To the best of our knowledge, this is the first time a RL system successfully learns how to automatically design BBO optimizers.

**Out-of-distribution Generalization.** For learning-assisted optimization techniques, the problem shifts in realistic scenarios might challenge their generalization ability in practice. To this end, we

test DesignX and MetaBBO baselines trained on synthetic $\mathcal{D}_{train}$ on three diverse realistic BBO testsuites: a) Protein-Docking [79], a collection of 280 protein-docking instances, featured by intricate landscapes; b) HPO-B [80], which comprises 86 ill-conditioning AutoML instances; c) UAV [81], 56 diverse conflict-free UAV path planning scenarios featured by implicit constraints multiplier in objective space (see Appendix E.3 for detail). We illustrate in Figure 3 the average optimization curves of all baselines, which is averaged within each tag and across 51 independent runs. The results show that: 1) DesignX generally shows superior optimization behavior to human-crafted optimizers from different decades, designing desirable optimizers robustly for diverse realistic problems it never saw during training; 2) DesignX consistently outperforms MetaBBO approaches, which demonstrates the novelty of our proposed bi-agent algorithm design system. By integrating two RL agents for both algorithmic workflow generation and hyper-parameter control, DesignX achieves better superior generalization performance to those MetaBBO baselines for single sub-task.

## 4.2   What has DesignX Learned?

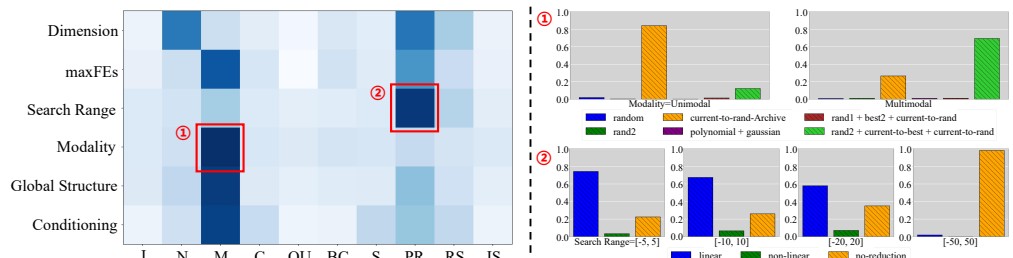

Figure 4: **Left:** Normalized importance factors of different module types for various problem characteristics. **Right:** Two look-into cases for interpreting design pattern learned by DesignX.

**Insightful Design Skills (RQ2).** Before delving into the analysis, we first abbreviate the 10 module types in Modular-EC to simplify the presentation: INITIALIZATION ("I"), NICHING ("N"), MUTATION ("M"), CROSSOVER ("C"), OTHER_UPDATE ("OU"), BOUNDARY_CONTROL ("BC"), SELECTION ("S"), POPULATION_REDUCTION ("PR"), RESTART_STRATEGY ("RS") and INFORMATION_SHARING ("IS"). The following analysis aims to investigate design principles DesignX has learned based on statistics gathered from the optimizer workflows generated for the 3200 problem instances in $\mathcal{D}_{test}$. We list several key observations we found as below:

1) In the left of Figure 4, we summarize the relative importance of different module types in Modular-EC when considering various optimization problem characteristics: Dimension, maxFEs, Search Range, Modality, Global Structure, Conditioning. To compute the relative importance, we provide a example here. Suppose we consider the relative importance of "M" (mutation) for Modality, we first divide problem instances in $\mathcal{D}_{test}$ into those unimodal ones and those multimodal ones. Then based on the optimizer workflows generated by DesignX for these problem instances, the relative importance can be calculated as the KL-divergence of the sub-module occurence distributions of "M" in unimodal problems and multimodal problems (see Appendix E.4 for more clarification). The relative importance factor reflects how DesignX thinks when designing an optimizer for a problem with certain property. As shown in Figure 4: a) for problems with different modalities, DesignX leans to design different DE mutation strategies for the generated workflow; b) for problem with different search ranges, DesignX leans to focus more on the selection of "PR" (population reduction mechanism). c) DesignX thinks designing initialization strategies has very limited impact on the final performance! These unique findings are non-trivial and deserve further analysis.

2) To investigate the above novel design principles interpreted from DesignX, we further look into the concrete sub-module occurence distributions in the first two cases. We illustrate them in the right of Figure 4. The results could clearly demonstrate DesignX's intelligent design policy: a) for unimodal problem, it smartly choose greedy-fashion mutation operators to reinforce the optimizer's exploitation, and dictates a composite mutation strategy for multimodal problems to address exploration and exploitation tradeoff. b) population reduction is an effective mechanism to upgrade an optimizer's local search ability. DesignX thinks for problems with relatively smaller searching range, population reduction should be applied to accelerate the convergence. c) we examine the finding of DesignX on Initialization by replacing the designs in existing optimizers with different ones. The results validate the correctness of DesignX and is shown in Appendix F.1.

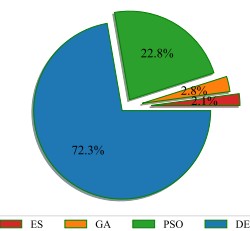

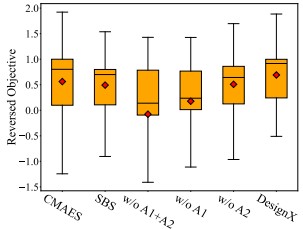

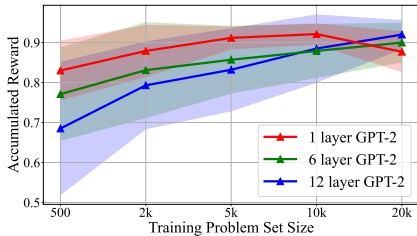

Figure 5: Ratios of selected module types.

Figure 6: Averaged performance of ablation baselines.

Figure 7: Performance comparison across model sizes and training sizes.

3) Another interesting design principle of DesignX is its unique taste on different optimizer types (DE, PSO, GA, ES). To illustrate this, we count the number of optimizers generated by DesignX which contain module variants derived from these four optimizer types, and then present their distribution in Figure 5. The results indicate that the DE-related algorithm sub-modules is primarily considered by DesignX to achieve aforementioned robust optimization performance. We provide several novel and very competitive DE optimizers discovered by DesignX in Appendix F.2.

## 4.3 In-depth Analysis

**Ablation Study (RQ3).** DesignX automates BBO optimizer design through the cooperation between Agent-1 and Agent-2. We hence investigate to what extent the two agents contribute to DesignX's final performance. Concretely, we introduce three ablations: 1) w/o A1+A2: randomized Agent-1 & 2 without training; 2) w/o A1: only Agent-2 is trained; and 3) w/o A2: only Agent-1 is trained. We also include two additional baselines for systematic analysis: 1) SBS: retrain DesignX without instance/observation features (set as zero vector), making Agent-1 learn a static algorithm workflow; 2) CMA-ES. We present the reversed normalized objective values (higher is better) of the ablations and DesignX on $\mathcal{D}_{test}$ and three realistic problem sets in Figure 6. Detailed results for each problem set are provided in Appendix F.3. The results reveal following insights: 1) we could at least conclude that generating a correct optimizer workflow might be more important than controlling the hyperparameters (w/o A2 v.s. w/o A1); 2) By training DesignX via our proposed cooperative learning objective, it achieves better performance than sub-task agent, which further validates the effectiveness of our method. 3) The static optimizer workflow found by SBS cannot perform very well, validating the necessity of generating customized workflows for each problem.

**Scaling Law (RQ4).** We further investigate the scalability of DesignX in terms of model capacity and training data scale. Due to our limited computational resources, a preliminary study is conducted here. Specifically, we investigate three different model sizes: 1,6 and 12 layers GPT-2 blocks for both Agent-1 and Agent-2, and five training problem set sizes: 500, 2000, 5000, 10000 and 20000. We train DesignX under the corresponding 15 combinations and report their testing performance on $\mathcal{D}_{test}$ in Figure 7. y-axis denoted the average learning objective across all tested problem instances and 51 independent runs. In general, we observe that when problem set scale is small, lager model might encounter overfitting issues hence underperforms on unseen problems. In contrast, for training-instance-rich scenario, larger model's learning ability continuously scales, while smaller ones might suffer from low capacity. However, in practice, it might consumes exponentially more resources for stable training in large models and training scales, hence in this paper, we select DesignX with 1 layer and 10k training scale as the final model. We additionally provide a comparison of DesignX and popular LLMs in terms of their design ability in Appendix F.4.

## 4.4 Additional Discussion

**Position of DesignX.** We discuss the differences of our DesignX with several related AAD fields: 1) Learning to Optimize (L2O) [82]: L2O commonly addresses gradient-based optimization, while MetaBBO such as DesignX addresses gradient-free ones; 2) Programming by Optimisation (PbO) [83]: PbO is tailored for a specific target scenario while DesignX aims at generalization ability for diverse optimization problems; 3) Combined Algorithm Selection and Hyperparameter optimization (CASH) [84]: CASH is a more general AutoML concept which focuses on selecting ML algorithms and determining static algorithm configuration. DesignX is capable of "creating" new ones and provides DAC capability.

Table 2: The detailed breakdown of DesignX's training time per training step.

| Agent | Agent-1 (7.5k steps) | | | | Agent-2 (2.2M steps) | | | |
|---|---|---|---|---|---|---|---|---|
| Process | Problem feature computation | Workflow generation | BBO process | Learning update | Optimization progress feature computation | Parameter values inference | BBO process | Learning update |
| Runtime | 2s | 0.95s | 20.01s | 0.03s | 0.001s | 0.02s | 0.04s | 0.09s |

**Computational Overhead.** We present a running time decomposition analysis of DesignX's training in Table 2, where the per-learning-step running time for each sub-process in the two agents' training is provided. To summarize, the true computational bottleneck of DesignX is not its deep learning framework, instead, is the inherent simulation cost in BBO optimization loop. This is also one of the major reason we train DesignX with CPUs, since BBO optimizers comprises many sequential logics that can not benefit from GPU's matrix and parallel acceleration support. Besides, if we put agents on GPU and the BBO process in CPU, the CPU-GPU communication overhead is also too much. When being used to solve a problem instance, DesignX consumes 5.5s on average and existing advanced optimizer such as CMA-ES consumes 5.0s, which indicates our DesignX would not increase the actual inference overhead. We also note that DesignX automatically "design" algorithm for users. Considering for a user with limited optimization knowledge, this could save much more time and lower the development difficulty.

**Rank-based Comparison.** In our major comparison experiment (Table 1), to showcase the evolution path of our DesignX during its days of training and also to provide an intuitive perspective on its generalization performance under knotty problem distribution, we use average performance as a major evaluation factor to show that DesignX has superior performance to those advanced optimizers in history. The conclusion may vary a little bit if we concentrate on the rank-based performance significance testing among all baselines. In Figure 8, we illustrate such perspective, where the critical difference diagram based on

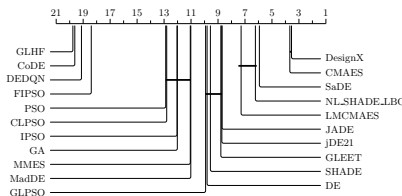

Figure 8: The rank-based comparison.

the same rollout data of Table 1 is provided. We can observe that when considering relative ranks across baselines, CMA-ES and DesignX performs almost equally and outperform the others significantly. This on the one hand demonstrates DesignX has learned robust policy, on the other hand, also outlines further improvement is still required.

**Potential Limitations.** While DesignX serves as an effective end-to-end learning framework for AAD, it holds certain limitations and anticipates more future research efforts. In this paper, the primary optimization domain is restricted within single-objective optimization. However, optimization domains such as multi-objective optimization requires more complex algorithm modules and corresponding training problems, which challenges the extensibility of DesignX. Thanks to the uniform interfaces we have developed in Modular-EC system, complex modules in such optimization domain can be integrated, and life-long learning/continual learning techniques might be a promising way to make DesignX embrace diverse optimization domains. Besides, we have to note that the training efficiency of DesignX is not very ideal. It still takes us 6 days to train DesignX on high-performance PC. Further improvement such as Multi-GPU parallel mechanism, proximal training objective, architecture simplification/distillation could be regarded as important future works.

## 5 Conclusion

In this paper, we propose DesignX as the first end-to-end MetaBBO approach which presents human-competitive end-to-end designing ability for BBO problems. We propose a novel dual-agent system with two RL agents for optimizer workflow generation and hyper-parameters control respectively. To effectively train DesignX, we construct a large scale synthetic problem set with 10k optimization problems with diverse characteristics. A cooperative learning objective is used to harmoniously learn optimal design policies for the two RL agents. Surprisingly, a DesignX model with merely two simple GPT-2 blocks continuously surpass popular human-crafted designs along the training. We have validated the generalization ability of DesignX on both synthetic and challenging realistic scenarios. More importantly, non-trivial design principles learned by DesignX are interpreted, which provides valuable design insights back to the community. We believe DesignX could serve as a pivotal step towards fully end-to-end foundation models for automated algorithm design.

## Acknowledgments and Disclosure of Funding

This work was supported in part by National Natural Science Foundation of China (Grant No. 62276100), in part by Guangzhou Science and Technology Elite Talent Leading Program for Basic and Applied Basic Research (Grant No. SL2024A04J01361), in part by the Guangdong Provincial Natural Science Foundation for Outstanding Youth Team Project (Grant No. 2024B1515040010), and in part by the Fundamental Research Funds for the Central Universities (Grant No. 2025ZYGXZR027).

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

# A Modular-EC

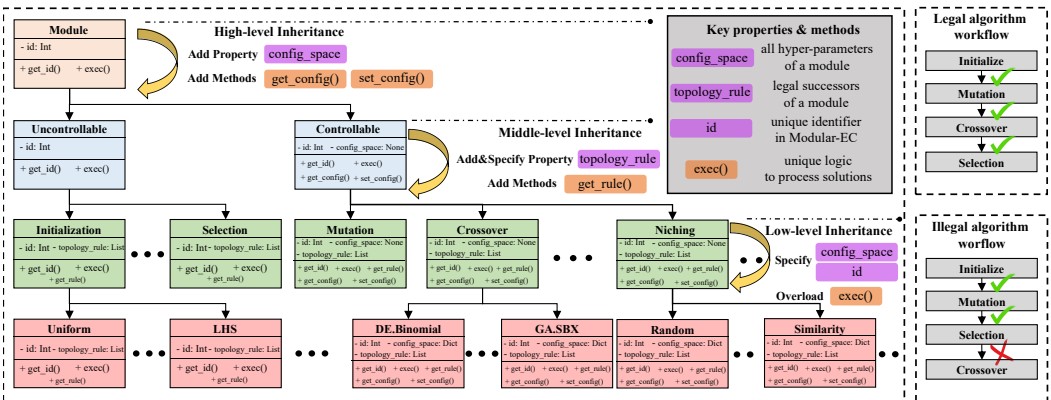

Figure 9: **Left**: The hierarchical *Python* inheritance in Modular-EC to support intricate polymorphism in EC modules. **Right**: Legal/Illegal algorithm workflow examples in Modular-EC.

**Hierarchical Inheritance.** As illustrated in the left of Figure 9, Modular-EC is designed as a Polymorphism system via multiple levels of *Python* inheritance. Such design allows maintaining diverse EC modules (the bottom ones in Figure 9) via universal interface encapsulation. In specific, Modular-EC holds three levels of inheritances: **1) High-level:** All modules in Modular-EC stem from the abstract base class MODULE, which declares properties and methods shared by all modules. In high-level inheritance, two sub-classes inherit from MODULE: UNCONTROLLABLE and CONTROLLABLE. These two sub-classes divide all possible EC modules into those without/with hyper-parameters. For CONTROLLABLE, we add a *config_space* property as its hyper-parameter space, which for now is void until a concrete EC module is created at the low-level inheritance; **2) Middle-level:** We have summarized several major EC module types from existing literature, which are widely adopted in many EC optimizers. In this inheritance level, UNCONTROLLABLE and CONTROLLABLE are further divided into these EC module types. Considering that a legal (or rational) EC optimizer workflow should comprises correctly ordered modules, we add and specify a *topology_rule* property for each module type to indicate which module types could be placed right after it. *topology_rule* plays a key role in DesignX's dual-agent algorithm design system to ensure legal generation of optimizer workflow in auto-regressive fashion. **3) Low-level:** In low-level inheritance, the concrete variants of each EC module types are created, which are collected by us from existing EC literature where they serve as common choices for many EC optimizers. For a concrete low-level module variant, we assign it a unique *id* property as its identifier in Modular-EC, specify *config_space* as a dictionary of its all hyper-parameters (if it inherits from CONTROLLABLE), and re-write *exec()* method by how it processes the solutions during optimization.

**Summary of Maintained EC Modules.** There are 6 UNCONTROLLABLE module types without hyper-parameters grouped in Modular-EC:

1. **INITIALIZATION** [45], which initialize a population of solutions to kick start a EC optimizer. We have maintained 5 initialization variants in the low-level inheritance (e.g., Sobol sampling [94], LHS sampling [95]).

2. **NICHING** [48], which divides the population into several sub-populations. We have maintained 3 niching variants in the low-level inheritance (Random [96], Ranking [97] and Distance [98]).

3. **BOUNDARY_CONTROL** [46], which ensures that the values of solutions in the population are all controlled in the bound. We have maintained 5 boundary control variants in the low-level inheritance (e.g., Clip [46], Reflect [46]).

4. **SELECTION** [47], which selects better solutions from parents/offsprings. We have maintained 6 variants of this type in the low-level inheritance (e.g., DE-Crowding [76], GA-Roulette [5]).

Table 3: The list of the practical variants of CONTROLLABLE and UNCONTROLLABLE modules.

(a) The CONTROLLABLE modules.

| type | Sub-module | | | |
|---|---|---|---|---|
| | Name + Id | Functional Description | Configuration Space | Topology Rule |
| MUTATION | DE/rand/1 [6]
1 - 000001 - 000000001 | Generate solution $x_i$'s trail solution $v_i = x_{r1} + F1 \cdot (x_{r2} - x_{r3})$
where $x_r$ are randomly selected solutions. | $F1 \in [0,1]$, default to 0.5. | Legal followers: DE-style CROSSOVER |
| | DE/rand/2 [6]
1 - 000001 - 000000010 | Generate solution $x_i$'s trail solution by $v_i = x_{r1} + F1 \cdot (x_{r2} - x_{r3}) + F2 \cdot (x_{r4} - x_{r5})$
where $x_r$ are randomly selected solutions. | $F1, F2 \in [0,1]$, default to 0.5. | Legal followers: DE-style CROSSOVER |
| | DE/best/1 [6]
1 - 000001 - 000000011 | Generate solution $x_i$'s trail solution by $v_i = x_{best} + F1 \cdot (x_{r1} - x_{r2})$
where $x_r$ are randomly selected solutions and $x_{best}$ is the best solution. | $F1 \in [0,1]$, default to 0.5. | Legal followers: DE-style CROSSOVER |
| | DE/best/2 [6]
1 - 000001 - 000000100 | Generate solution $x_i$'s trail solution by $v_i = x_{best} + F1 \cdot (x_{r1} - x_{r2}) + F2 \cdot (x_{r3} - x_{r4})$
where $x_r$ are randomly selected solutions and $x_{best}$ is the best solution. | $F1, F2 \in [0,1]$, default to 0.5. | Legal followers: DE-style CROSSOVER |
| | DE/current-to-best/1 [6]
1 - 000001 - 000000101 | Generate solution $x_i$'s trail solution by $v_i = x_i + F1 \cdot (x_{best} - x_i) + F2 \cdot (x_{r1} - x_{r2})$
where $x_r$ are randomly selected solutions and $x_{best}$ is the best solution. | $F1, F2 \in [0,1]$, default to 0.5. | Legal followers: DE-style CROSSOVER |
| | DE/current-to-rand/1 [6]
1 - 000001 - 000000110 | Generate solution $x_i$'s trail solution by $v_i = x_i + F1 \cdot (x_{r1} - x_i) + F2 \cdot (x_{r2} - x_{r3})$
where $x_r$ are randomly selected solutions. | $F1, F2 \in [0,1]$, default to 0.5. | Legal followers: DE-style CROSSOVER |
| | DE/rand-to-best/1 [6]
1 - 000001 - 000000111 | Generate solution $x_i$'s trail solution by $v_i = x_{r1} + F1 \cdot (x_{best} - x_{r2})$
where $x_r$ are randomly selected solutions and $x_{best}$ is the best solution. | $F1 \in [0,1]$, default to 0.5. | Legal followers: DE-style CROSSOVER |
| | DE/current-to-pbest/1 [69]
1 - 000001 - 000001000 | Generate solution $x_i$'s trail solution by $v_i = x_i + F1 \cdot (x_{pbest} - x_i) + F2 \cdot (x_{r1} - x_{r2})$
where $x_r$ are randomly selected solutions and $x_{pbest}$ is a randomly selected from the top $p$ best solutions. | $F1, F2 \in [0,1]$, default to 0.5;
$p \in [0,1]$, default to 0.05. | Legal followers: DE-style CROSSOVER |
| | DE/current-to-pbest/1+archive [69]
1 - 000001 - 000001001 | Generate solution $x_i$'s trail solution by $v_i = x_i + F1 \cdot (x_{pbest} - x_i) + F2 \cdot (x_{r1} - x_{r2})$
where $x_{r1}$ is a randomly selected solutions, $x_{r2}$ is randomly selected from the union of the population and the archive which contains inferior solutions, $x_{pbest}$ is a randomly selected solution from the top $p$ best solutions. | $F1, F2 \in [0,1]$, default to 0.5;
$p \in [0,1]$, default to 0.05. | Legal followers: DE-style CROSSOVER |
| | DE/weighted-rand-to-pbest/1 [75]
1 - 000001 - 000001010 | Generate solution $x_i$'s trail solution by $v_i = F1 \cdot x_{r1} + F1 \cdot F2 \cdot (x_{pbest} - x_{r2})$
where $x_r$ are randomly selected solutions and $x_{best}$ is the best solution. | $F1, F2 \in [0,1]$, default to 0.5;
$p \in [0,1]$, default to 0.05. | Legal followers: DE-style CROSSOVER |
| | DE/current-to-rand/1+archive [75]
1 - 000001 - 000001011 | Generate solution $x_i$'s trail solution by $v_i = x_i + F1 \cdot (x_{r1} - x_i) + F2 \cdot (x_{r2} - x_{r3})$
where $x_{r1}, x_{r2}$ are randomly selected solutions, $x_{r3}$ is randomly selected from the union of the population and the archive which contains inferior solutions. | $F1, F2 \in [0,1]$, default to 0.5. | Legal followers: DE-style CROSSOVER |
| | Gaussian_mutation [5]
1 - 000001 - 000001100 | Generate a mutated solution of $x_i$ by adding a Gaussian noise on each dimension
$v_i = \mathcal{N}(x_i, \sigma \cdot (ub - lb))$ where $ub$ and $lb$ are the upper and lower bounds of the search space. | $\sigma \in [0,1]$, default to 0.1 | Legal followers: BOUNDARY_CONTROL |
| | Polynomial_mutation [85]
1 - 000001 - 000001101 | Generate a mutated solution of $x_i$ as $v_i = \begin{cases} x_i + ((2u)^{\frac{1}{1+\eta_m}} - 1)(x_i - lb), \text{if } u \leq 0.5; \\ x_i + (1 - (2 - 2u)^{\frac{1}{1+\eta_m}})(ub - x_i), \text{if } u > 0.5. \end{cases}$
where $u \in [0,1]$ is a random number, $ub$ and $lb$ are the upper and lower bounds of the search space. | $\eta_m \in [20,100]$, default to 20 | Legal followers: BOUNDARY_CONTROL |
| | Multi_Mutation_1 [75]
1 - 000100 - 000000001 | Contains DE/current-to-pbest/1+archive, DE/current-to-rand/1+archive and DE/weighted-rand-to-best/1 three DE mutation sub-modules, its first configuration is to select one of the three mutations and the rest configurations are to configured the selected operator. | $op \in \{$DE/CURRENT-TO-PBEST/1+ARCHIVE, DE/CURRENT-TO-RAND/1+ARCHIVE, DE/WEIGHTED-RAND-TO-BEST/1$\}$, random selection in default;
$F1, F2 \in [0,1]$, default to 0.5;
$p \in [0,1]$, default to 0.18. | Legal followers: DE-style CROSSOVER |
| | Multi_Mutation_2 [70]
1 - 000100 - 000000010 | Contains DE/rand/1, DE/rand/2 and DE/current-to-rand/1 three DE mutation sub-modules. | $op \in \{$DE/RAND/1, DE/RAND/2, DE/CURRENT-TO-RAND/1$\}$, random selection in default;
$F1, F2 \in [0,1]$, default to 0.5; | Legal followers: DE-style CROSSOVER |
| | Multi_Mutation_3 [99]
1 - 000100 - 000000011 | Contains DE/rand/1, DE/best/2 and DE/current-to-rand/1 three DE mutation sub-modules. | $op \in \{$DE/RAND/1, DE/BEST/2, DE/CURRENT-TO-RAND/1$\}$, random selection in default;
$F1, F2 \in [0,1]$, default to 0.5; | Legal followers: DE-style CROSSOVER |
| | 33 more Mutation Multi-Strategies are omitted here since they are too many to presenting them one by one.
1 - 000100 - 000000100
~1 - 000100 - 000110001 | ... | ... | ... |
| CROSSOVER | Binomial [6]
1 - 000010 - 000000001 | Randomly exchange values between parent solution $x_i$ and the trail solution $v_i$ to get a new solution:
$u_{i,j} = \begin{cases} v_{i,j}, \text{if } rand_j < Cr \text{ or } j = jrand \\ x_{i,j}, \text{otherwise} \end{cases}, j = 1, \cdots, D$ where $rand_j \in [0,1]$ is a
random number, $jrand \in [1, D]$ is a randomly selected index before crossover and $D$ is the solution dimension. | $Cr \in [0,1]$, default to 0.9. | Legal followers: BOUNDARY_CONTROL |
| | Exponential [6]
1 - 000010 - 000000010 | Exchange a random solution segment between $x_i$ and $v_i$ to get a new solution:
$u_{i,j} = \begin{cases} v_{i,j}, \text{if } rand_{k,j} < Cr \text{ and } k \leq j \leq L + k \\ x_{i,j}, \text{otherwise} \end{cases}, j = 1, \cdots, D$ where $k \in [1, D]$ is a randomly
selected start point for exchanging, $L \in [1, D - k]$ is a randomly determined exchange length, $rand_{k,j} \in [0,1]^{D-k}$ is the random numbers from index $k$ to $j$ and $D$ is the solution dimension. | $Cr \in [0,1]$, default to 0.9. | Legal followers: BOUNDARY_CONTROL |
| | qbest_Binomial [87]
1 - 000010 - 000000011 | Randomly exchange values between a solution $x_i^t$ selected from the top $p$ population and the trail solution $v_i$ to get a new solution:
$u_{i,j} = \begin{cases} v_{i,j}, \text{if } rand_j < Cr \text{ or } j = jrand \\ x_{i,j}^t, \text{otherwise} \end{cases}, j = 1, \cdots, D$ where $rand_j \in [0,1]$ is a
random number, $jrand \in [1, D]$ is a randomly selected index before crossover and $D$ is the solution dimension. | $Cr \in [0,1]$, default to 0.9;
$p \in [0,1]$, default to 0.5 | Legal followers: BOUNDARY_CONTROL |
| | qbest_Binomial+archive [75]
1 - 000010 - 000000100 | Randomly exchange values between a solution $x_i^t$ selected from the top $p$ population-archive union and the trail solution $v_i$ to get a new solution:
$u_{i,j} = \begin{cases} v_{i,j}, \text{if } rand_j < Cr \text{ or } j = jrand \\ x_{i,j}^t, \text{otherwise} \end{cases}, j = 1, \cdots, D$ where $rand_j \in [0,1]$ is a
random number, $jrand \in [1, D]$ is a randomly selected index before crossover and $D$ is the solution dimension. | $Cr \in [0,1]$, default to 0.9;
$p \in [0,1]$, default to 0.18 | Legal followers: BOUNDARY_CONTROL |
| | SBX [88]
1 - 000010 - 000000101 | Generate child solution(s) $v_i$ by $v_i = 0.5 \cdot [(1 \mp \beta)x_{p1} + (1 \pm \beta)x_{p2}]$
where $\beta = \begin{cases} (2u)^{\frac{1}{\eta_c+1}} - 1, \text{if } u \leq 0.5; \\ (\frac{1}{2-2u})^{\frac{1}{\eta_c+1}}, \text{if } u > 0.5. \end{cases}, u \in [0,1]$ is a random number, $x_{p1}$ and $x_{p2}$ are two
randomly selected parents. | $\eta_c \in [20,100]$, default to 20 | Legal followers: GA-style MUTATION |
| | Arithmetic [89]
1 - 000010 - 000000110 | Generate child solution $v_i$ by $v_i = (1 - \alpha) \cdot x_{p1} + \alpha \cdot x_{p2}$ where $x_{p1}$ and $x_{p2}$ are two
randomly selected parents. | $\alpha \in [0,1]$, default to 0.5. | Legal followers: GA-style MUTATION |
| | Multi_Crossover_1 [75]
1 - 000100 - 000110010 | Contains Binomial and qbest_Binomial+archive two DE crossover sub-modules. | $op \in \{$BINOMIAL, QBEST_BINOMIAL+ARCHIVE$\}$, random selection in default;
$Cr \in [0,1]$, default to 0.9; | Legal followers: BOUNDARY_CONTROL |
| | Multi_Crossover_2 [99]
1 - 000100 - 000110011 | Contains Binomial and Exponential two DE crossover sub-modules. | $op \in \{$BINOMIAL, EXPONENTIAL$\}$, random selection in default;
$Cr \in [0,1]$, default to 0.9; | Legal followers: BOUNDARY_CONTROL |
| | 9 more Crossover Multi-Strategies are omitted here since they are too many to presenting them one by one.
1 - 000100 - 000110100
~1 - 000100 - 000111101 | ... | ... | ... |
| OTHER_UPDATE | Vanilla_PSO [7]
1 - 000011 - 000000001 | Update solution $x_i^t$ at generation $t$ using $x_i^{t+1} = x_i^t + vel_i^t$ where velocity vector
$vel_i^t = w \cdot vel_i^{t-1} + c1 \cdot rand_1 \cdot (pbest_i^t - x_i^t) + c2 \cdot rand_2 \cdot (gbest^t - x_i^t)$,
$rand_\cdot \in (0,1]$ are random values, $pbest_i^t$ is the best solution $x_i$ ever achieved, $gbest^t$ is the global best solution. | $w \in [0.4, 0.9]$, default to 0.7;
$c1, c2 \in [0, 2]$, default to 1.49445. | Legal followers: BOUNDARY_CONTROL |
| | FDR_PSO [91]
1 - 000011 - 000000010 | Update solution $x_i^t$ at generation $t$ using $x_i^{t+1} = x_i^t + vel_i^t$ where velocity vector
$vel_i^t = w \cdot vel_i^{t-1} + c1 \cdot rand_1 \cdot (pbest_i^t - x_i^t) + c2 \cdot rand_2 \cdot (gbest^t - x_i^t) + c3 \cdot rand_3 \cdot (nbest_i^t - x_i^t)$,
$rand_\cdot \in (0,1]$ are random values, $pbest_i^t$ is the best solution $x_i$ ever achieved, $gbest^t$ is the global best solution and $nbest_i^t$ is the solution that maximizes the Fitness-Distance-Ratio $nbest_{i,j}^t = x_{p,j}^t$ which
$p_j = \arg \max \frac{f_i^t - f_p^t}{\mu_{p,j}^t - x_{i,j}^t}, j = 1, \cdots, D, f$ denotes the objective values and $D$ is solution dimension. | $w \in [0.4, 0.9]$, default to 0.729;
$c1, c2 \in [0, 2]$, default to 1;
$c3 \in [0, 2]$, default to 2. | Legal followers: BOUNDARY_CONTROL |
| | CLPSO [68]
1 - 000011 - 000000011 | Update solution $x_i^t$ at generation $t$ using $x_i^{t+1} = x_i^t + vel_i^t$ where velocity vector
$vel_i^t = w \cdot vel_i^{t-1} + c1 \cdot rand_1 \cdot (pbest_{f_i}^t - x_i^t) + c2 \cdot rand_2 \cdot (gbest^t - x_i^t)$, where $rand_\cdot \in (0,1]$ are
random values, $gbest^t$ is the global best solution, $pbest_{f_i,d}^t = \begin{cases} pbest_{i,j}^t, \text{if } rand_j > Pc_i; \\ pbest_{z,j}^t, \text{otherwise.} \end{cases}, j = 1, \cdots, D$
is the ever achieved best solution of $x_i$ or $x_z$ which is randomly selected with fitness based tournament. | $w \in [0.4, 0.9]$, default to 0.7;
$c1, c2 \in [0, 2]$, default to 1.49445. | Legal followers: BOUNDARY_CONTROL |
| | CMA-ES [65]
1 - 000011 - 000000100 | Given a population $x_t$ and the corresponding objective values $y_t$ at generation $t$, CMA-ES updates its Gaussian mean $\omega_t$, covariance matrix $C_t$, and global step size $\sigma_t$ following [65], then samples the next population $x_{t+1} \sim \mathcal{N}(\omega_t, \sigma_t^2 \cdot C_t)$ | $c \in [0.1, 1]$, default to 1;
$cs \in [0.1, 1]$, default to 1. | Legal followers: BOUNDARY_CONTROL |
| | Sep-CMA-ES [92]
1 - 000011 - 000000101 | Given a population $x_t$ and the corresponding objective values $y_t$ at generation $t$, Sep-CMA-ES updates its Gaussian mean $\omega_t$, diagonal elements for the covariance matrix $D_t$, and global step size $\sigma_t$ following [92], then samples the next population $x_{t+1} \sim \mathcal{N}(\omega_t, \sigma_t^2 \cdot D_t)$. | $c \in [0.1, 1]$, default to 1;
$cs \in [0.1, 1]$, default to 1. | Legal followers: BOUNDARY_CONTROL |
| | MMES [77]
1 - 000011 - 000000110 | By incorporating the Fast Mixture Sampling (FMS) [77] into a generic $(\mu, \lambda)$-ES, the next population is sampled by $x_{i+1}^t \sim \omega_t + \sigma_t \cdot z_i^t$ where $\omega_t$ is the Gaussian mean, $\sigma_t$ is the mutation strength, and $z_i^t$ is a mutation vector sampled by FMS. | $c \in [0.1, 1]$, default to 1;
$cs \in [0.1, 1]$, default to 1. | Legal followers: BOUNDARY_CONTROL |
| | Multi_PSO_1 [93]
1 - 000100 - 000001010 | Contains FDR_PSO and CLPSO two PSO update sub-modules. | $op \in \{$FDR_PSO, CLPSO$\}$, random selection in default;
$w \in [0.4, 0.9]$, default to 0.729;
$c1, c2 \in [0, 2]$, default to 1;
$c3 \in [0, 2]$, default to 2. | Legal followers: BOUNDARY_CONTROL |
| | 3 more Multi-Strategies about Other_Updates are omitted here since they are too many to presenting them one by one.
1 - 000100 - 000001011
~1 - 000100 - 000001101 | ... | ... | ... |
| INFORMATION_SHARING | Sharing [3]
1 - 000101 - 000000001 | Receive the best solution from the target sub-population and replace the worst solution in current sub-population. | $target \in [1, N_{nich}]$, random selection in default | Legal followers: POPULATION_REDUCTION, end |

5. **RESTART_STRATEGY** [49], which re-initializes the population when it converges or stagnates. We have maintained 4 restart strategy variants in the low-level inheritance (e.g., Stagnation [99], Obj_Convergence [76]).

6. **POPULATION_REDUCTION** [50], which reduces the population size to perform exploitative optimization. We have maintained 2 variants of this type in the low-level inheritance (Linear [100] and Non-Linear [101]).

For CONTROLLABLE modules, we introduce four types:

1. **MUTATION** [51], which introduces stochastic local search for each solution. We have maintained 49 mutation variants in the low-level inheritance (e.g., GA-gaussian [5], DE/rand/1 [6]).

(b) The UNCONTROLLABLE modules.

| type | Sub-module | | |
|---|---|---|---|
| | Name + Id | Functional Description | Topology Rule |
| INITIALIZATION | Uniform [45]
0 - 000001 - 000000001 | Uniformly sample solutions in the search range $x \sim \mathcal{U}(lb, ub)$
where $ub$ and $lb$ are the upper and lower bounds of the search space. | Legal followers: DE-style MUTATION, PSO_UPDATE, GA-style CROSSOVER |
| | Sobol [102]
0 - 000001 - 000000010 | Sample population in Sobol' sequences. | Legal followers: DE-style MUTATION, PSO_UPDATE, GA-style CROSSOVER |
| | LHS [95]
0 - 000001 - 000000011 | Sample population in Latin hypercube sampling. | Legal followers: DE-style MUTATION, PSO_UPDATE, GA-style CROSSOVER |
| | Halton [103]
0 - 000001 - 000000100 | Sample population in Halton sequence. | Legal followers: DE-style MUTATION, PSO_UPDATE, GA-style CROSSOVER |
| | Normal [104]
0 - 000001 - 000000101 | Sample solutions in Normal distribution $x \sim \mathcal{N}((ub + lb)/2, \frac{1}{6}(ub - lb))$
where $ub$ and $lb$ are the upper and lower bounds of the search space. | Legal followers: DE-style MUTATION, PSO_UPDATE, GA-style CROSSOVER |
| NICHING | Rand [96]
0 - 000010 - 000000001 | Randomly partition the overall population into $N_{nich} \in [2, 4]$ same size sub-populations. | Legal followers: DE-style MUTATION, PSO_UPDATE, GA-style CROSSOVER |
| | Ranking [97]
0 - 000010 - 000000010 | Sort the population according to their fitness and partition them into $N_{nich} \in [2, 4]$ same size sub-populations. | Legal followers: DE-style MUTATION, PSO_UPDATE, GA-style CROSSOVER |
| | Distance [98]
0 - 000010 - 000000011 | Randomly select a solution and assign its $NP//N_{nich} - 1$ nearest solutions to a new sub-population, until all solutions are assigned. | Legal followers: DE-style MUTATION, PSO_UPDATE, GA-style CROSSOVER |
| BOUNDARY_CONTROL | Clip [46]
0 - 000011 - 000000001 | Clip the solutions out of bounds at the bound $x_i = \text{clip}(x_i, lb, ub)$ | Legal followers: SELECTION |
| | Rand [46]
0 - 000011 - 000000010 | Randomly regenerate those out of bounds $x_{i,j} = \begin{cases} x_{i,j}, \text{ if } lb_j \leq x_{i,j} \leq ub_j, \\ \mathcal{U}(lb_j, ub_j), \text{ otherwise} \end{cases}$ | Legal followers: SELECTION |
| | Periodic [46]
0 - 000011 - 000000011 | Consider the search range as a closed loop
$x_{i,j} = \begin{cases} x_{i,j}, \text{ if } lb_j \leq x_{i,j} \leq ub_j, \\ lb_j + ((x_{i,j} - ub_j) \mod (ub_j - lb_j)), \text{ otherwise} \end{cases}$ | Legal followers: SELECTION |
| | Reflect [46]
0 - 000011 - 000000100 | Reflect the values that hit the bound $x_{i,j} = \begin{cases} 2ub_j - x_{i,j}, \text{ if } ub_j < x_{i,j}, \\ 2lb_j - x_{i,j}, \text{ if } x_{i,j} < lb_j, \\ x_{i,j}, \text{ otherwise} \end{cases}$ | Legal followers: SELECTION |
| | Halving [46]
0 - 000011 - 000000101 | Halve the distance between the $x_i$ and the crossed bound
$x_{i,j} = \begin{cases} x_{i,j} + 0.5 \cdot (x_{i,j} - ub_j), \text{ if } ub_j < x_{i,j}, \\ x_{i,j} + 0.5 \cdot (x_{i,j} - lb_j), \text{ if } x_{i,j} < lb_j, \\ x_{i,j}, \text{ otherwise} \end{cases}$ | Legal followers: SELECTION |
| SELECTION | DE-like [6]
0 - 000100 - 000000001 | Select the better one from the parent solution and its trail solution. | Legal followers: RESTART_STRATEGY, POPULATION_REDUCTION, end, INFORMATION_SHARING (If NICHING is used) |
| | Crowding [76]
0 - 000100 - 000000010 | The trail solution complete against its closest solution and the better one survives. | Legal followers: RESTART_STRATEGY, POPULATION_REDUCTION, end, INFORMATION_SHARING (If NICHING is used) |
| | PSO-like [7]
0 - 000100 - 000000011 | Replace the old population with the new solutions without objective value comparisons. | Legal followers: RESTART_STRATEGY, POPULATION_REDUCTION, end, INFORMATION_SHARING (If NICHING is used) |
| | Ranking [105]
0 - 000100 - 000000100 | Select solutions for the next generation according to the ranking based probabilities, with the worst one ranking 1, the probability of the solution rank $i$ is $p_i = \frac{1}{NP}(p^+ + (p^+ - p^-)\frac{i-1}{NP-1})$ where $NP$ is the population size, $p^+$ is the probability of selecting the best solution and $p^-$ is the probability of selecting the worst one. | Legal followers: RESTART_STRATEGY, POPULATION_REDUCTION, end, INFORMATION_SHARING (If NICHING is used) |
| | Tournament [106]
0 - 000100 - 000000101 | Randomly pair solutions and select the better one in each pair for the next generation. | Legal followers: RESTART_STRATEGY, POPULATION_REDUCTION, end, INFORMATION_SHARING (If NICHING is used) |
| | Roulette [5]
0 - 000100 - 000000110 | Select solutions according to the fitness based probabilities $p_i = \frac{f_i'}{\sum_{j=1}^{NP} f_j'}$ where $f_j'$ is the fitness of the $j$-th solution and $NP$ is population size. | Legal followers: RESTART_STRATEGY, POPULATION_REDUCTION, end, INFORMATION_SHARING (If NICHING is used) |
| RESTART_STRATEGY | Stagnation [99]
0 - 000101 - 000000001 | Reinitialize the population if the improvement of the best objective value is equal to or less than a threshold $10^{-10}$ for 100 generations. | Legal followers: end |
| | Obj_Convergence [76]
0 - 000101 - 000000010 | Reinitialize the population if the maximal difference of the objective values of the top 20% solutions is less than a threshold $10^{-16}$. | Legal followers: end |
| | Solution_Convergence [107]
0 - 000101 - 000000011 | Reinitialize the population if the maximal difference of the solutions on all dimensions are less than a threshold $10^{-16}$ search space diameter. | Legal followers: end |
| | Obj&Solution_Convergence [108]
0 - 000101 - 000000100 | Reinitialize the population if the maximal difference of the objective values is less than threshold $10^{-8}$ and the maximal distance among solutions is less than 0.005 search space diameter. | Legal followers: end |
| POPULATION_REDUCTION | Linear [100]
0 - 000110 - 000000001 | Linearly reduce the population size from the initial size $NP_{max}$ to the minimal population size $NP_{min}$. The size at generation $g + 1$ is $NP_{g+1} = round((NP_{min} - NP_{max}) \cdot \frac{g}{H}) + NP_{max}$ where $g$ is the generation number and $H$ is the optimization horizon. | Legal followers: Restart_Strategy, end |
| | Non-Linear [101]
0 - 000110 - 000000010 | Non-linearly determine the $g + 1$ generation population size as $NP_{g+1} = round((NP_{min} - NP_{max})^{1-g/H} + NP_{max})$ where $NP_{min}$ and $NP_{max}$ are the minimal and maximal population sizes, $g$ is the generation number and $H$ is the optimization horizon. | Legal followers: Restart_Strategy, end |
| end | end
0 - 000111 - 000000001 | A token indicating the completion of algorithm structure generation which has no practical function. | – |

2. **CROSSOVER** [52], which encourages global optimization by exchanging two solution's information. We have maintained 17 crossover variants in the low-level inheritance (e.g., GA-SBX [5], DE-binomial [6]).

3. **OTHER_UPDATE**, which denotes other population update paradigm in PSO/ES variants. We have maintained 10 variants of this type in the low-level inheritance (e.g., ES-CMA [65], ES-Diagonal [92], PSO-normal [7]).

4. **INFORMATION_SHARING** [53], which takes the best solution in the target sub-population to replace the worst solution in current sub-population to perform information sharing between sub-populations.

Additionally, advanced evolutionary computation (EC) methods often integrate multiple candidate operators to dynamically select operators during optimization. To accommodate such scenarios, we introduce MULTI_STRATEGY modules, which contains 2-5 candidate sub-modules of the same type (e.g., mutation operators) and expose an additional operator selection parameter in their configuration space (*config_space*). For categorization, Multi-Strategy modules inherit the type of their constituent sub-modules. For example, a MULTI_STRATEGY module containing DE mutation operators is classified under the MUTATION category.

**Module's ID.** The unique identifier *id* of a module variant is a 16-bit binary code of which: 1) the first bit is 0 or 1 to denote if this variant is UNCONTROLLABLE or CONTROLLABLE; 2) the 2-nd to

7-th bits denote which one of the 11 module types this variant belongs to; 3) the last 9 bits denotes its id within that module type.

**Module's Topology Rule.** A key property in a module variant is its *topology_rule*, which is a list of module types indicating which module types are allowed to be placed right after this module variant. A very simple example is illustrated in the right of Figure 9, where in a EC optimizer, selection modules are not allowed to be placed before crossover modules. We list some other examples here: 1) Any EC optimizer must start with INITIALIZATION; 2) BOUNDARY_CONTROL is not allowed placed between two subsequent reproduction modules (e.g., MUTATION and CROSSOVER); 3) RESTART_STRATEGY is only allowed to be placed at the end of a EC optimizer.

In total, we have created 116 module variants in the low-level inheritance to cover commonly used techniques in existing EC literature. Besides, an *end* token is included to indicate the end of the algorithm generation. We provide a complete information table about these module variants in Table 3a and Table 3b, including their names, types, original papers and hyper-parameters (*config_space*). Such a comprehensive module space in Modular-EC could express BBO optimizers with diverse workflow structures, hence allows learning for effective (even optimal) algorithm design policies.

# B  Feature Design

## B.1  ELA Features for Agent-1

In this paper for each problem we introduce a 13-dimensional feature vector $\mathcal{F}_p$ comprising two components: the 4-dimensional basic problem information and the 9-dimensional ELA features. The basic information includes the problem dimension ($D$), maxFEs ($maxFEs$), upper bound ($ub$) and lower bound ($lb$). Since the scale of these values could vary, we normalize the feature of problem dimension $\mathcal{F}_D = \frac{1}{5}\log_{10} D$ and the feature of maxFEs $\mathcal{F}_{FEs} = \frac{1}{10}\log_{10} maxFEs$. For the upper and lower bounds we use $\mathcal{F}_{ub} = ub/100$ and $\mathcal{F}_{lb} = lb/100$ respectively. For the 9-dimensional ELA features which are significant independence and efficient computation according to the sensitivity analysis of ELA features in [59, 60], we present them in Table 4. These features profile the optimization properties of the problem such as modality, Skewness, global structures, etc.

Table 4: The list of the ELA features for Agent-1.

| Features | Description |
| --- | --- |
| ela_meta.lin_simple.intercept | The intercept of the linear regression model approximating the problem. |
| ela_meta.quad_simple.adj_r2 | Adjusted coefficient of determination of the quadratic regression model without variable interactions. |
| ela_meta.lin_w_interact.adj_r2 | Adjusted coefficient of determination of the linear regression model with variable interactions. |
| ic.m0 | The initial partial information from the Information Content of Fitness Sequences (ICoFiS) approach [109]. |
| ic.h_max | The maximum information content from ICoFiS. |
| ic.eps_ratio | The half partial information sensitivity from ICoFiS. |
| nbc.nn_nb.mean_ratio | The ratio of arithmetic mean based on the distances among the nearest neighbors and the nearest better neighbors. |
| nbc.dist_ratio.coeff_var | The coefficient of variation of the distance ratios. |
| ela_distr.number_of_peaks | The estimation of the number of peaks in the distribution of the function values. |

## B.2  Statistical Features for Agent-2

The statistical feature $\mathcal{O}_t \in \mathbb{R}^9$ is summarized below:

1. The first feature is the minimum objective value in the current (sub-)population indicating the achieved best performance of the current (sub-)population:

$$\mathcal{O}_{i,1} = \min\{\frac{f_i}{f^{0,*} - f^*}\}_{i \in [1, NP_{local}]} \tag{8}$$

   It is normalized by the difference between the best objective value at initial optimization $f^{0,*}$ step and the global optimal objective value of the optimization problem $f^*$, so that the scales of the features from different tasks are in the same level. which hence stabilizes the training. $NP_{local}$ is the (sub-)population size.

2. The second one is the averaged normalized objective values in the current (sub-)population, indicating the average performance of the (sub-)population:

$$\mathcal{O}_{i,2} = \text{mean}\{\frac{f_i}{f^{0,*} - f^*}\}_{i \in [1, NP_{local}]} \tag{9}$$

3. The variance of the normalized objective values in the current (sub-)population, indicating the variance and convergence of the (sub-)population:

$$\mathcal{O}_{i,3} = \text{std}\{\frac{f_i}{f^{0,*} - f^*}\}_{i \in [1, NP_{local}]} \tag{10}$$

4. The next feature is the maximal distance between the solutions in (sub-)population, normalized by the diameter of the search space, measuring the convergence:

$$\mathcal{O}_{i,4} = \max_{i,j \in [1, NP_{local}]} \frac{||x_i - x_j||_2}{||ub - lb||_2} \tag{11}$$

where $ub$ and $lb$ are the upper and lower bounds of the search space.

5. The dispersion difference [110] feature is calculated as the difference of the maximal distance between the top 10% solutions and the maximal distance between all solutions in (sub-)population:

$$\mathcal{O}_{i,5} = \max_{i,j \in [1, 10\% NP_{local}]} \frac{||x_i - x_j||_2}{||ub - lb||_2} \\ - \max_{i,j \in [1, NP_{local}]} \frac{||x_i - x_j||_2}{||ub - lb||_2} \tag{12}$$

It measures the funnelity of the problem landscape: a single funnel problem has a smaller dispersion difference while the multi-funnel landscape has larger value.

6. The fitness distance correlation (FDC) [111] describes the complexity of the problem by evaluating the relationship between fitness value and the distance of the solution from the optimum.

$$\mathcal{O}_{i,6} = \frac{\frac{1}{NP_{local}} \sum_{i=1}^{NP_{local}} (f_i - \bar{f})(d_i^* - \bar{d}^*)}{\text{var}(\{d_i^*\}_{i \in [1, NP_{local}]}) \cdot \text{var}(\{f_i\}_{i \in [1, NP_{local}]})} \tag{13}$$

where the $\bar{f}$ is the averaged objective value in (sub-)population, $d_i^* = ||x_i - x^*||_2$ is the distance between $x_i$ and the best solution $x^*$, $\bar{d}^* = \text{mean}\{d_i^*\}_{i \in [1, NP_{local}]}$ is the averaged distance,
var(·) is the variance.

7. The found global best objective among all (sub-)populations, indicating the achieved best performance of the overall optimization:

$$\mathcal{O}_{i,7} = \min\{\frac{f_i}{f^{0,*} - f^*}\}_{i \in [1, NP]} \tag{14}$$

8. This feature is the FDC feature for the overall population:

$$\mathcal{O}_{i,8} = \frac{\frac{1}{NP} \sum_{i=1}^{NP} (f_i - \bar{f})(d_i^* - \bar{d}^*)}{\text{var}(\{d_i^*\}_{i \in [1, NP]}) \cdot \text{var}(\{f_i\}_{i \in [1, NP]})} \tag{15}$$

9. The last feature is the remaining optimization budget, indicating the optimization progress:

$$\mathcal{O}_{i,9} = \frac{maxFEs - FEs}{maxFEs} \tag{16}$$

where $maxFEs$ is maximum allowed function evaluations and $FEs$ is the number of consumed function evaluations.

## C  Synthetic Problem Set Generation

To construct the large scale synthetic problem set, we first collect 32 representative basic problem functions from popular benchmarks [61, 62], which are listed in Table 5. Given a solution $x \in \mathbb{R}^D$, a shift vector $o \in \mathbb{R}^D$ and a rotation matrix $M \in \mathbb{R}^{D \times D}$, the objective value of a $D$-dimensional basic problem with problem function $f_b$ is formulated as $F_b(x) = f_b(M^T(x - o))$. Then to enhance problem diversity, we borrow the idea from CEC benchmarks [61] and construct the "composition" and "hybrid" problems.

Table 5: Overview of the basic problem functions.

| ID | Functions | Modality | Global Structure | Conditioning |
|---|---|---|---|---|
| $f_1$ | Sphere | Unimodal | Adequate | Low |
| $f_2$ | Schwefel F12 | Unimodal | Adequate | Low |
| $f_3$ | Ellipsoidal | Unimodal | Adequate | Low |
| $f_4$ | Ellipsoidal high condition | Unimodal | Adequate | High |
| $f_5$ | Bent cigar | Unimodal | Adequate | High |
| $f_6$ | Discus | Unimodal | Adequate | High |
| $f_7$ | Different Powers | Unimodal | Adequate | High |
| $f_8$ | Rosenbrock | Unimodal | Adequate | Low |
| $f_9$ | Ackley | Multimodal | Adequate | High |
| $f_{10}$ | Weierstrass | Multimodal | Weak | High |
| $f_{11}$ | Griewank | Multimodal | Weak | Low |
| $f_{12}$ | Rastrigin | Multimodal | Weak | High |
| $f_{13}$ | Buche-Rastrigin | Unimodal | Adequate | High |
| $f_{14}$ | Modified Schwefel | Multimodal | Weak | High |
| $f_{15}$ | Katsuura | Multimodal | Weak | High |
| $f_{16}$ | Composite Griewank-Rosenbrock Function F8F2 | Unimodal | Adequate | Low |
| $f_{17}$ | Escaffer's F6 | Multimodal | Adequate | High |
| $f_{18}$ | Happycat | Multimodal | Weak | Low |
| $f_{19}$ | Hgbat | Unimodal | Adequate | Low |
| $f_{20}$ | Lunacek bi-Rastrigin | Multimodal | Weak | High |
| $f_{21}$ | Zakharov | Unimodal | Adequate | Low |
| $f_{22}$ | Levy | Multimodal | Weak | High |
| $f_{23}$ | Scaffer's F7 | Multimodal | Weak | Low |
| $f_{24}$ | Step-Rastrigin | Multimodal | Weak | Low |
| $f_{25}$ | Linear Slope | Unimodal | Adequate | Low |
| $f_{26}$ | Attractive Sector | Unimodal | Adequate | High |
| $f_{27}$ | Step-Ellipsoidal | Multimodal | Weak | Low |
| $f_{28}$ | Sharp Ridge | Unimodal | Adequate | High |
| $f_{29}$ | Rastrigin's F15 | Unimodal | Weak | Low |
| $f_{30}$ | Schwefel | Multimodal | Weak | Low |
| $f_{31}$ | Gallagher's Gaussian 101-me Peaks | Multimodal | Weak | Low |
| $f_{32}$ | Gallagher's Gaussian 21-hi Peaks | Multimodal | Weak | Low |

"composition" problems aggregate basic problems using weighted sum. It first randomly select $n$ basic problem functions as the sub-problems $\{f^1, \cdots, f^n\}$ where $n \in [2, 5]$. Then for the $i$-th sub-problem we generate a weight $w^i \in (0, 1]$. Finally, the composition problem $F_c$ is calculated as the weighted sum of objective values of its sub-problems $F_c(x) = \sum_{i=1}^{n} w^i f^i(M^{\mathrm{T}}(x - o))$ where $x$ is the solution, $o$ is the shift vector and $M$ is the rotation matrix.

"hybrid" problems decomposition solutions into several segments and evaluate these segments with different sub-problems. It first randomly decomposes $D$ problem dimensions into $n \in [2, 5]$ segments with each segment $s^i = \{d^{i,0}, \cdots, d^{i,D^i}\}$ where $d^{i,j} \in [1, D]$ is the index of the $j$-th dimension in the segment, $D^i$ is the length of the $i$-th segment satisfying $\sum_{i=1}^{n} D^i = D$. Then $n$ basic problem functions are selected as the sub-problems $\{f^1, \cdots, f^n\}$ with dimensions $\{D^1, \cdots, D^n\}$ respectively. The evaluation of hybrid problem $F_h$ is defined as $F_h(x) = \sum_{i=1}^{n} f^i((M^{\mathrm{T}}(x - o))[s^i])$.

To construct the 12800 problem instances, for each problem, the problem type is randomly selected from "single" (basic problem), "composition" and "hybrid". The problem dimension is chosen from $\{5, 10, 20, 50\}$, the search range is sampled from $\{[-5, 5], [-10, 10], [-20, 20], [-50, 50]\}$ and the maxFEs is selected from $\{10000, 20000, 30000, 40000, 50000\}$. If the problem type is "single", its problem function is randomly selected from the 32 basic problem functions. If the problem type is "composition" or "hybrid", 2-5 sub-problems as well as their weights or dimension decompositions are determined. After the construction of 12800 problems, we then randomly split them into a training problem set $\mathcal{D}_{train}$ with 9600 problems and a testing problem set $\mathcal{D}_{test}$ with 3200 problems.

# D  Pseudo Code of Training

The cooperative training of DesignX is two-stage. Started by three initial models, the Agent-1 model $\pi_\phi$, Agent-2 actor $\pi_\theta$ and critic $v_\psi$, we firstly train Agent-1 and freeze Agent-2 models. For each epoch and each problem $p \in \mathcal{D}_{train}$ with dimension $D$, $100 \cdot D$ solutions are sampled, evaluated and then used to calculate the ELA features $\mathcal{F}_{ELA}$ of problem $p$. Given the feature vector $\mathcal{F}_p$ concatenated by basic problem information and $\mathcal{F}_{ELA}$, Agent-1 auto-regressively generates the modules $\mathcal{A}_p$ using

$\mathcal{F}_p$ as mentioned in Section 3.2.1 in the main paper. Controlled by the frozen Agent-2, $\mathcal{A}_p$ optimizes problem $p$ using $p.maxFEs$ function evaluations and obtains the accumulated reward $R_p$, which is then used to update $\pi_\phi$ in REINFORCE [63] manner. After training Agent-1, the well-trained model is frozen and its Agent-2's turn. For each epoch and each problem $p \in \mathcal{D}_{train}$, Agent-1 generates an effective algorithm with modules $\mathcal{A}_p$. For each optimization step, the Agent-2 actor $\pi_\theta$ determines the hyper-parameters of the CONTROLLABLE modules in $\mathcal{A}_p$ according to the current state $\mathcal{O}_t$. The controlled $\mathcal{A}_p$ optimizes $p$ for one step and obtains the next state $\mathcal{O}_{t+1}$ and reward $r_t$. For each $nstep$ optimization, the actor $\pi_\theta$ and critic $v_\psi$ are updated for $kepoch$ learning steps in a PPO [64] manner. The pseudo code is shown in Alg. 1. We omit the batch processing for better readability.

---

**Algorithm 1:** The pseudo code of the training of DesignX

---

**Input:** Training problem set $\mathcal{D}_{train}$, Modular-EC $\mathcal{M}$, initial Agent-1 model $\pi_\phi$, Agent-2 actor $\pi_\theta$ and critic $v_\psi$.
**Output:** Well trained $\pi_\phi$, $\pi_\theta$ and $v_\psi$.
// Training for Agent-1;
Freeze $\pi_\theta$;
**for** $epoch = 1$ **to** $Epoch$ **do**
    **for** *each* $p \in \mathcal{D}_{train}$ **do**
        Sample solutions $X_{ELA} \in \mathbb{R}^{100p.D \times p.D}$ and evaluate them $Y_{ELA} = p(X_{ELA})$;
        Obtain the ELA features $\mathcal{F}_{ELA} = \text{ELA}(X_{ELA}, Y_{ELA})$;
        Get the feature vector $\mathcal{F}_p = \text{Concat}(p.D, p.maxFEs, p.ub, p.lb, \mathcal{F}_{ELA})$;
        Auto-regressively generate the optimizer $\mathcal{A}_p = \pi_\phi(\mathcal{F}_p, \mathcal{M})$ following Section 3.2.1;
        Initial state $\mathcal{O}_{t=1} = \mathcal{A}_p.optimize(p)$, $R_p = 0$;
        **while** *Termination condition is not met* **do**
            $a_t = \pi_\theta(\mathcal{O}_t)$;
            $\mathcal{O}_{t+1}, r_t = \mathcal{A}_p.optimize(a_t, p)$;
            $R_p = R_p + r_t$;
        **end**
        Update $\pi_\phi$ by in REINFORCE [63] manner;
    **end**
**end**
// Training for Agent-2;
Freeze $\pi_\phi$;
**for** $epoch = 1$ **to** $Epoch$ **do**
    **for** *each* $p \in \mathcal{D}_{train}$ **do**
        Generate the optimizer $\mathcal{A}_p$ as Lines 7~10;
        Initial state $\mathcal{O}_{t=1} = \mathcal{A}_p.optimize(p)$;
        **while** *Termination condition is not met* **do**
            **for** $step = 1$ **to** $nstep$ **do**
                $a_t = \pi_\theta(\mathcal{O}_t)$;
                $\mathcal{O}_{t+1}, r_t = \mathcal{A}_p.optimize(a_t, p)$;
                Record transition $< s_t, a_t, s_{t+1}, r_t >$;
            **end**
            **for** $k = 1$ **to** $kepoch$ **do**
                Update actor $\pi_\theta$ and critic $v_\psi$ in PPO [64] manner;
            **end**
         **end**
    **end**
**end**

---

# E Experimental Setup

## E.1 Training Setup

In this paper, we set the embedding dimension $h = 64$ and the number of attention head $k = 4$ for both Agent-1 & 2. The number of blocks $L$ is 1 for Agent-1 and 3 for Agent-2. The maximum number

of modules $M$ is 64 and the predefined maximum configuration size $N_{max} = 12$. The training of both agents on $\mathcal{D}_{train}$ lasts for $Epoch = 100$ epochs with a fixed learning rate 0.0001. Agent-1 is trained with a batch size of 128. During the training of Agent-2, for a batch of 64 problems, PPO [64] method is used to update the policy and critic nets for $kepoch = 3$ times for every $nstep = 10$ rollout optimization steps. All experiments are run on two Intel(R) Xeon(R) 6458Q CPUs with 488G RAM. All baseline configurations align with their original papers.

## E.2 Objective Value Normalization

Since the objective value scales of different problems can vary, averaging them directly is not fair, it cannot reflect the true performance of baselines. To normalize the values to the same scale, we use the best objective value found by random search $f^*_{p,RS}$ on problem $p$. Concretely, for problem $p$ we randomly sample $p.maxFEs$ solutions in the search range $[p.lb, p.ub]$ and take the best sampled objective value as $f^*_{p,RS}$. In the experiment, for the found best objective value $f^*_{p,b,i}$ of baseline $b$ in test run $i$ on problem $p$, we normalize it by $f^{*\prime}_{p,b,i} = \frac{f^*_{p,b,i}}{f^*_{p,RS}}$. Then we average the normalized objective values of baseline $b$ on all problem and all runs as the normalized averaged objective value in Table 1 in the main paper: $f_b = \frac{1}{|\mathcal{D}_{test}| \cdot 51} \sum_{p \in \mathcal{D}_{test}} \sum_{i=1}^{51} f^{*\prime}_{p,b,i}$. The similar procedure is conducted on the three realistic benchmarks. We also use a reversed normalized averaged objective value formulated as $1 - f_b$ in the ablation study in Section 4.3.

## E.3 Realistic Benchmark

1. **Protein-Docking Benchmark** [79], where the objective is to minimize the Gibbs free energy resulting from protein-protein interaction between a given complex and any other conformation. We select 28 protein complexes and randomly initialize 10 starting points for each complex, resulting in 280 problem instances. To simplify the problem structure, we only optimize 12 interaction points in a complex instance (12D problem).

2. **HPO-B Benchmark** [80] is an AutoML hyper-parameter optimization benchmark which includes a wide range of hyperparameter optimization tasks for 16 different model types (e.g., SVM, XGBoost, etc.), resulting in a total of 935 problem instances. The dimension of these problem instances range from 2 to 16. To save evaluation time, we adopt the continuous version of HPO-B, which provides surrogate evaluation functions for time-consuming machine learning tasks. We also note that HPO-B represents problems with ill-conditioned landscape such as huge flatten.

3. **UAV Path Planning Benchmark** [81] provides 56 terrain-based landscapes as realistic Unmanned Aerial Vehicle (UAV) path planning problems, each of which is 30D. The objective is to select given number of path nodes (x,y,z coordinates) from the 3D space, so the the UAV could fly as shortly as possible in a collision-free way.

## E.4 Relative Importance Calculation

Taking the relative importance of mutation ("M") modules on modality as an example, we first divide problem instances in $\mathcal{D}_{test}$ into those unimodal ones and those multimodal ones. Next we collect the mutation modules used in optimizers generated for unimodal and multimodal problems respectively. We count the occurence of each mutation sub-modules in the two mutation module collections as the histogram shown in the top right of Figure 4 in the main paper. Considering the occurence probabilities of different sub-modules in the two collections for unimodal and multimodal as two distributions, we then measure the relative importance of mutation modules to modality as the KL-divergence between the two distributions. For characteristics with more than two properties such as dimension, maxFEs and search range, we use the maximum KL-divergence among the distributions. Finally, to highlight the relative importance of different modules to the same problem characteristic, we conduct the mean-std standardization. Given the importance $\mathcal{I}_{\omega,\rho}$ of module $\omega \in \Omega$ to characteristic $\rho$, the standardized importance is $\mathcal{I}'_{\omega,\rho} = \frac{\mathcal{I}_{\omega,\rho} - \text{mean}_{\varpi \in \Omega}(\mathcal{I}_{\varpi,\rho})}{\text{std}_{\varpi \in \Omega}(\mathcal{I}_{\varpi,\rho})}$, which is shown in the left of Figure 4 in the main paper.

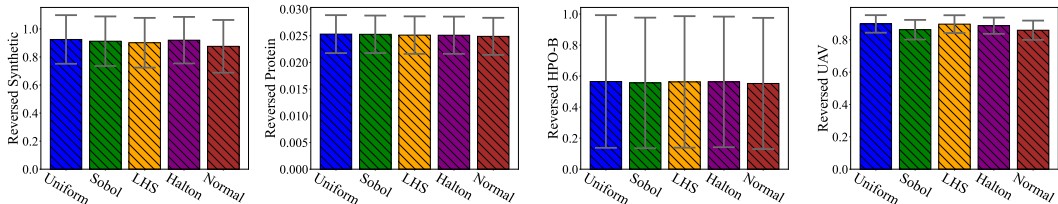

Figure 10: The performance of optimizers with 5 different initialization modules on $\mathcal{D}_{test}$ and three realistic benchmarks.

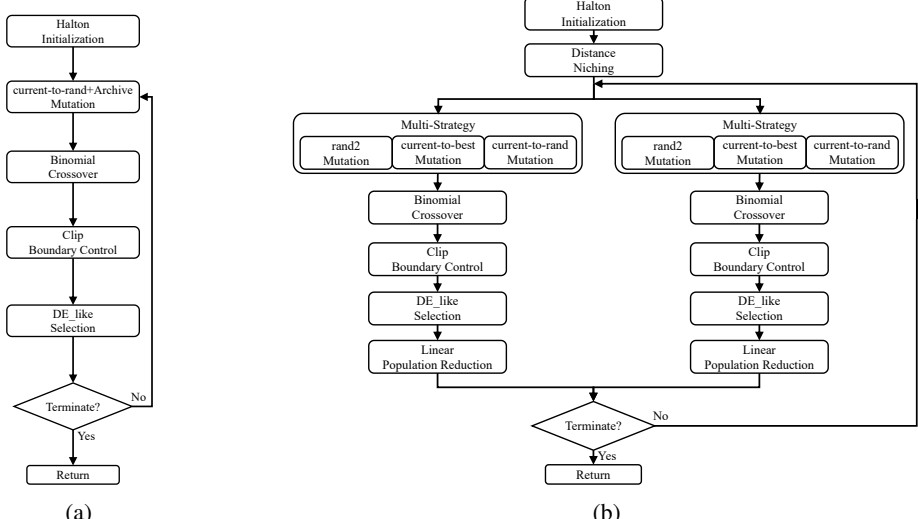

Figure 11: Two examples of DesignX generated DE optimizers.

## F  Additional Experimental Results

### F.1  Insightful Design Skills in Initialization

In Section 4.2 of the main paper, we observed that certain modules (e.g., Initialization) contribute minimally to optimizer performance. To validate this finding, we replace the Initialization modules in existing optimizers with five sampling methods: Uniform sampling [45], Sobol sampling [94], Latin-hypercube sampling (LHS) [95], Halton sampling [103] and Normal sampling [104]. The performance of optimizers with different Initialization modules on $\mathcal{D}_{test}$ and three realistic benchmarks are demonstrated in Figure 10. The results show that different Initialization modules have limited impact on the optimization performance, which validates the correctness of DesignX: The influence of different initialization methods might be diminished by subsequence more important optimization modules such as mutation modules.

### F.2  Examples of Generated DE Optimizers

In this section we provide two examples of the competitive DE optimizers discovered by DesignX in Figure 11. Figure 11a is a simple DE/current-to-rand/1/binomial optimizer with an archive for eliminated individuals. It could perform efficiency exploitative optimization on unimodal problems. Figure 11b is a relatively complex DE optimizer with two sub-populations split by a Distance-based Niching module which enhances the population diversity. The two sub-populations both use a mutation multi-strategy module containing 3 DE mutations: rand2, current-to-rand and current-to-best, followed by the Binomial crossover. The composite mutation modules not only address exploration and exploitation tradeoff but also provide Agent-2 more decision flexibility. Besides, linear population reduction modules are introduced to accelerate the convergence at the end of optimization. These designs make the optimizer superior in solving multimodal problems. The two examples validate the intelligence and effectiveness of DesignX.

## F.3 Additional Results of Ablation baselines

In this section we demonstrate the detailed ablation study results for $\mathcal{D}_{test}$ and the three realistic benchmarks in Figure 12. The results validate that generating optimizer workflow (w/o A2) is more important than hyper-parameter control (w/o A1) in general cases. On the other hand, it is quite obvious that training Agent-1 and Agent-2 in a cooperative way results in better optimization performance. We also observe that the ablated models and the final DesignX model perform equally in HPO-B tasks, this might reveal that the generalization of DesignX on extremely ill-conditioned BBO scenarios is still limited. This might be addressed by some RL-based fine-tuning on specifically constructed ill-conditioned problem set.

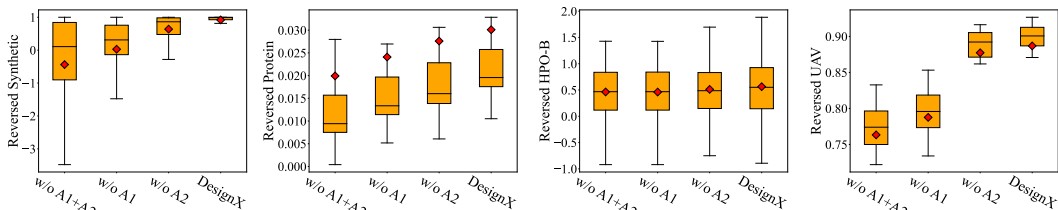

Figure 12: Detailed performance of ablation baselines on $\mathcal{D}_{test}$ and three realistic benchmarks.

Table 6: Normalized averaged performance of DesignX and LLMs on synthetic and realistic problems.

|                    | GPT-4 Turbo | Gemini-1.5 | Deepseek-R1 | DesignX |
|--------------------|-------------|------------|-------------|---------|
| $\mathcal{D}_{test}$ | 2.21E-01    | 2.08E-01   | 2.31E-01    | **8.26E-02** |
|                    | ±7.68E-02   | ±1.22E-01  | ±7.26E-02   | **±1.75E-01** |
| Protein Docking    | 9.72E-01    | 9.72E-01   | 9.71E-01    | **9.69E-01** |
|                    | ±2.57E-06   | ±2.44E-06  | ±2.51E-06   | **±2.43E-06** |
| HPO-B              | 3.78E-01    | 3.95E-01   | 4.36E-01    | **3.44E-01** |
|                    | ±1.89E-02   | ±2.10E-02  | ±1.98E-02   | **±1.85E-02** |
| UAV                | 1.28E-01    | 1.31E-01   | 1.25E-01    | **1.17E-01** |
|                    | ±1.20E-02   | ±1.79E-02  | ±1.23E-02   | **±2.30E-02** |

## F.4 Comparison to LLMs

We would like to note that Large Language Models (LLMs) is also capable of designing algorithms for diverse tasks [18, 19]. In the context of Optimization, however, the potential and expertise level of existing general LLMs may not be very ideal. To demonstrate this, in this section, we consider three LLM baselines: GPT-4 Turbo [112], Gemini-1.5 [113] and Deepseek-R1 [114], and compare their algorithm design ability with our DesignX model on $\mathcal{D}_{test}$ and three realistic problem sets. For each tested problem instance we prompt the LLMs with a design requirement: *"You are an expert in Black-Box Optimization, given a problem instance with following mathematical form: xxx, and given its dimension as 10D, search range as [-10, 10], optimization budget as 10000 function evaluations. Please generate an optimizer with executable code for this problem. Do not generate explanations!"*. Then we execute their generated optimizer code to optimize the problem. The averaged results are shown in Table 6. DesignX significantly outperforms LLMs across all benchmarks. While LLMs is demonstrated with powerful general-task-solving capability, the results here clearly indicate their lacks of optimization-domain-specific knowledge. By checking the codes these LLMs generated, we found that these general LLMs are only capable of recognizing current task is an optimization task, while ignoring the specific problem characteristics behind. A direct demonstration is that they lean to generate a specific kind of optimizer: Vanilla DE, for almost all tested problem instances. In contrast, DesignX is trained specifically to tailor desired optimizers for diverse optimization problems. Through its learning from Modular-EC, valuable expert-level knowledge from human experts are effectively injected into the two agents. The cooperative large-scale training enables DesignX's Agent-1 and Agent-2 learn optimal workflow generation policy and parameter control policy respectively, resulting in state-of-the-art performance.

