# OpenReview forum: "DesignX: Human-Competitive Algorithm Designer for Black-Box Optimization"
_NeurIPS.cc/2025/Conference — NeurIPS 2025 poster_

### Official Review · Reviewer_WYcX · 2025-06-30

**Clarity:** 3
**Significance:** 2
**Originality:** 4
**Rating:** 4
**Confidence:** 5

**Summary:**

The authors propose DesignX a novel meta-learning method for the design of model-free black box optimizers. The method consists of two agents, both using transformer policies whose weights are jointly optimized using reinforcement learning on a set of synthetic optimization tasks. The first agent is used at optimization start to select the modular algorithmic workflow conditioned on characteristics of the problem (per-instance algorithm selection) and the second agent is used during every step of the optimization to dynamically determine the parameter values for each module (dynamic algorithm configuration). Authors evaluate the meta-learned reactive black box optimizer on in-distribution and out-of-distribution tasks, and present additional analyses and an ablation regarding the relative contributions of both agents.

**Questions:**

I have serious concerns that in my view render the manuscript as-is unsuitable for publishing. That being said, if all these concerns are fully addressed, I am open to considerably increase my evaluation score and “accept” the paper.

- Improve positioning in historical AAD context. MetaBBO is a very recent movement that mainly relabels and unifies research directions that have been explored in the algorithm selection, configuration and hyper-heuristic community for decades. Also more recent trends like “learning to optimize” (https://www.jmlr.org/papers/volume23/21-0308/21-0308.pdf) are closely related. I would like to ask the authors to discuss its relation to these lines of works. Furthermore, since agent 2 “automatically learns a policy for hyperparameter tuning”, Dynamic Algorithm Configuration  (DAC, https://www.jair.org/index.php/jair/article/view/13922/26882) should be mentioned.
- Nuance / clarify the claim that DesignX is “the first framework that jointly learns workflow generation and tuning”. In particular, they should relate this contribution to previous work in “programming by optimization” and on the “CASH” problem.
- Improve the presentation of the results in Table 1 and Figure 3. In particular, I strongly feel $\textit{the authors should not average results across multiple methods}$. In addition, I recommend to complement the results with rank-based statistics, ideally combined with rank-based statistical significant tests / critical difference diagrams (https://mirkobunse.github.io/CriticalDifferenceDiagrams.jl/stable/), which would give a better impression of relative performance across the heterogeneous sets. For example in Table 1, instead of picking 20 arbitrary instances, one could partition test instances based on their properties (e.g., dimensionality / modality / etc.) and compute aggregated rank-based statistics per subgroup. Similarly, in Figure 3, results are averaged across many different instances, which is susceptible to outliers and including average rank plots is best practice in these settings. Specifically for Figure 3, using a log scale for the x-axis would be more appropriate, since function evaluations (at least in AutoML) are expensive.
- Include a baseline (e.g., CMA-ES) to put the existing ablation results into perspective. Also, if possible, include the single-best-solver (SBS) as an ablation on Agent 1. This could be done by retraining DesignX without instance/observation features, causing it to effectively learn a single algorithm template and static setting. Alternatively, a static algorithm configurator (e.g., SMAC) could be used on the Modular-EC search space.
- Discuss limitations (see Limitations part of this review).

**Ethical Concerns:**

["NO or VERY MINOR ethics concerns only"]

**Final Justification:**

The authors' prompt and elaborate responses have addressed most of my original concerns and assuming the promised changes are implemented, I do not see further ground for rejection. I have decided to improve my score to borderline accept (4). For a better score, I am missing a clear / consistent improvement w.r.t. e.g. CMA-ES which, given the complexity of the approach, I believe one could reasonably expect. Nonetheless, the approach is novel and interesting and further work along these lines could bring future improvements.

**Limitations:**

The authors do not discuss the limitations of DesignX.


In particular, I would like to see a discussion of these aspects of the method / evaluation:
- Computational overhead / resource requirements: Hand-crafted optimizers, in particular evolutionary methods typically add relatively little overhead per Function Evaluation (FE). The overhead of DesignX is likely much higher, as it must query Agent 2 every iteration. The comparison in Figure 3 (#FE on x-axis) is therefore not entirely fair. Furthermore, while most evolutionary methods run of CPU, it is unclear whether DesignX requires GPU.
- Conceptual complexity / tunability: DesignX is relatively complicated. An important benefit of evolutionary / model-free optimization methods is that they tend to be simple, and relatively easy to tune and adjust when needed. Doing the same for DesignX seems hardly trivial.
- Evolutionary BBO: DesignX is limited to evolutionary methods. These are not state-of-the-art across all scenarios. In particular, they are known to require a lot of FEs compared to more sample-efficient, model-based optimizers such as Bayesian Optimization.
In settings were FEs are expensive, e.g. AutoML, one would typically use Bayesian Optimization methods to get near-optimal results in 10-100 FEs.
- Tuning of hand-crafted baselines: It is unclear how the parameters for the hand-crafted baselines were chosen. Decades of academic literature and practice have demonstrated the importance of optimizing these for the task at hand. Comparing a method that was meta-learned for 6 days against baselines with default hyperparameter setting is arguably unfair.

**Paper Formatting Concerns:**

No major issues, some typos I came across:

- line 44 optimzier
- line 177/178: ... let Agent 2 aware ... (broken sentence)
- matrix $\mathcal{W}_{token}$ in 3.2.1 is defined twice with different dimensions (13 x h and 16 x h)

**Quality:**

2

**Strengths And Weaknesses:**

$\textbf{Strengths:}$ The paper presents an interesting and novel approach to automated algorithm design (AAD). To the best of my knowledge, this is the first work to combine per-instance algorithm selection with dynamic algorithm configuration in an end-to-end fashion. The method is described in sufficient detail and design decisions explained and motivated. While not the first line of work in this direction, I think the use of deep-RL with modern sequence models (in this case transformers) is very promising. The evaluation, in particular the meta-learning curve shown in Figure 1, as well as the ablations, show this meta-learning approach works, at least to some extent.

$\textbf{Weaknesses:}$ The authors make some very strong claims (e.g., “the DesignX-generated optimizers continuously surpass human-crafted optimizers by orders of magnitude”), which combined with a lacking positioning in the historical context of AAD, the absence of a critical discussion of limitations, and questionable presentation of the main result, is prone to misinterpretations (especially given the diversity of the NeurIPS target audience).

 - While I regard the approach taken in this work to be interesting and novel, I am somewhat concerned about how the authors position their contribution. The related work section, despite claiming to “review the development of Automated Algorithm Design (AAD) over the past decades”  jumps from Genetic programming to recent works published in the last 5 years, ignoring over four decades of active research on the topic.
- The main novelty claim made is that DesignX is “the first framework that jointly learns workflow generation and tuning”.
  - First, the distinction between workflow / tuning is not that clear in AAD, parameters often affect workflow and workflow components can be seen as categorical parameters. For example, formulating the AAD as an optimization / configuration problem has been explored under the name programming by optimization (PbO: https://www.cs.ubc.ca/~hoos/Publ/Hoos10.pdf, early examples: https://www.ijcai.org/Proceedings/09/Papers/093.pdf, https://citeseerx.ist.psu.edu/document?repid=rep1&type=pdf&doi=fa23d5b7d494fe40bde7417e3639c7cbd7f347ac, http://www.cmap.polytechnique.fr/~nikolaus.hansen/proceedings/2014/WCCI/CEC-2014/PROGRAM/E-14713.pdf).
  - Second, while the classical works distinguish between algorithm selection/configuration, the combined Algorithm Selection and Hyperparameter Optimization (CASH problem) has been extensively studied (https://dl.acm.org/doi/10.1145/2487575.2487629).
- My primary concern is about how the main results are presented. Specifically, I feel $\textit{the presentation of results in Table 1 and Figure 3 is misleading}$: The averaging of the results of all baselines in the same decade is problematic. One poor member in the generation can easily pull down an entire generation - especially since averages are reported which are susceptible to outliers. For example, inspecting the data provided by the authors, on the first instance in Table 1, CMA-ES has 4.27E-05 (6.02E-06) while designX obtains 2.89E-01. Orders of magnitude worse (LM-CMA-ES even obtains 5.28E-12). Also, showing 20 random instances in Table 1 is not the best way of summarizing the results across the 3200 test functions and the use of rank-based statistics aggregated across all instances would be more appropriate (see Questions to authors), also in Figure 3.
- The ablations, while interesting, are also very strong (replacing learned by random choices). For Agent 2 (DAC), it is known (and confirmed) that random can be a strong baseline, but for agent 1 (AS), improvement over random algorithm selection is arguably setting the bar too low. In particular, it is best practice in AS to include a comparison with the single best solver (SBS, see also Questions to authors).
- Failure to discuss limitations. I do not follow the justification by the authors: “The limitations of the work are not discussed due to page limitation”. A lack of space should not be the reason not to discuss limitations. While the main limitations should arguably be mentioned in the main paper, they could have at least added an Appendix discussing these.

---

> ### Author Rebuttal · Authors · 2025-07-30
>
> We want to express our deepest gratitude to the reviewer for recognizing our paper as a novel and interesting work. Besides, we also thank you for the valuable and insightful comments. For your concerns, we provide following point-to-point responses to address them.
>
> **[W1&W2&Q1&Q2, Positioning and Related Works Discussion]**
>
> We appreciate for the constructive suggestions and agree that key works in related areas should be replenished to enhance the paper’s scientific integrity. To this end, we provide following discussion.
>
> 1. L2O:  Sharing the same meta learning paradigm, key difference between L2O and MetaBBO locates at the targeted optimization domain: L2O commonly addresses gradient-based optimization problems (so called White-Box), while MetaBBO such as DesignX addresses gradient-free (black-box) scenarios.
> 2. PbO: While level-0 to level-3 PbO is closely related to some automated algorithm design concepts such as parameter selection, configuration, operator selection, algorithm selection, etc. DesignX shows great similarity with the highest level PbO (level-4): providing designs for every where of a program/workflow. However, two differences remain: a) PbO requires context-based program specification for a specific target scenario, which requires developer to provide these prototypes as much as possible, while in contrast, DesignX provides abstract modular space that includes generic operators adaptable for a wide range of scenarios. b) PbO focuses on improving performance on specific tasks, while DesignX serves as a pivotal step toward foundation model with generalization ability for diverse optimization problems.
> 3. CASH: CASH methods have been proven effective in many AutoML scenarios, such as selecting machine learning algorithms and corresponding hyper-parameter values for specific tasks. In comparison, DesignX’s target scenarios are a subset of the CASH target scenarios, designing BBO optimizers. However, in terms of methodology, DesignX shows more flexibility in the design process: a) CASH focuses on selecting existing algorithms while DesignX is capable of “creating” new ones. b) CASH determines a parameter setting once for all, while DesignX supports adaptive control.
> 4. Others: We also have to note that dynamic algorithm configuration, algorithm selection and hyper-heuristic are also closely related to DesignX. Thanks to the long-standing study on them, DesignX draws valuable insights from them and achieves automated end-to-end algorithm design.
>
> Considering PbO and CASH, we decide to change our claim from “the first framework that jointly learns workflow generation and tuning” to “the first deep learning framework that jointly learns workflow generation and tuning” in the revised paper.
>
> We hope the above discussion could address your concerns. We will refine such discussion into the related work of the revised paper.
>
> **[W3&Q3, Better Statistical Presentation]**
>
> The aggregated results shown in Table 1 and Figure 3 are a deliberate choice to manage the scale of our comparison. Evaluating DesignX against 20 BBO/MetaBBO baselines on thousands of problems generated an immense volume of data, making it impossible to present individual results for all baselines and problems within the confines of the paper. Table 1, therefore, reports average performance on a subset of 20 randomly selected problem instances. Full, disaggregated results have been provided in an Excel spreadsheet within the anonymous GitHub repository (since the time of submission).
>
> In addition, we appreciate the reviewer for this brilliant idea of using rank-based statistical diagrams!  Following your suggestions, we have conducted all possible partitions of the tested set. The suggested statistical validation is also conducted and the results show that CMA-ES and our DesignX dominate the other baselines with 95% confidence in all tested cases. We showcase two partitions and the corresponding ranking results of the top 10 baselines below and will add all results to the revised paper (Appendix).
>
> **Unimodal & Multimodal**
>
> |  | DE | CMAES | SaDE | JADE | SHADE | LMCMAES | jDE21 | NL-SHADE-LBC | GLEET | DesignX |
> | --- | --- | --- | --- | --- | --- | --- | --- | --- | --- | --- |
> | **Unimodal** | 10.513 | **3.850** | 5.392 | 8.743 | 9.117 | 5.518 | 8.783 | 7.332 | 9.901 | 4.276 |
> | **Multimodal** | 9.840 | 3.497 | 6.216 | 8.702 | 9.617 | 7.715 | 8.707 | 6.100 | 8.884 | **2.952** |
>
> **Adequate Global Structure & Weak Global Structure**
>
> |  | DE | CMAES | SaDE | JADE | SHADE | LMCMAES | jDE21 | NL-SHADE-LBC | GLEET | DesignX |
> | --- | --- | --- | --- | --- | --- | --- | --- | --- | --- | --- |
> | **Adequate** | 9.761 | **2.771** | 6.130 | 8.959 | 9.351 | 7.184 | 8.835 | 6.351 | 8.861 | 3.544 |
> | **Weak** | 10.880 | 4.870 | 6.136 | 7.156 | 10.830 | 9.353 | 7.980 | 5.473 | 8.884 | **3.640** |
>
> We can observe that: for easier problems such as those with unimodal or adequate global structure, CMA-ES achieves better performance due to its straightforward convergence. However, for those multimodal and weak global structure problems, our DesignX is capable of discovering more promising algorithms. We thank the reviewer for leading us to a promising way to present the advantages of our DesignX.
>
> For Figure 3, we will provide the same ranking results in Appendix and use a log scale for the x-axis in the revised paper.
>
> **[W4&Q4, Ablation Baselines]**
>
> We have added CMAES and the SBS as baselines according to your suggestion. We report the average performance of them in the following table.
>
> | CMAES | SBS | w/o A1+A2 | w/o A1 | w/o A2 | DesignX |
> | --- | --- | --- | --- | --- | --- |
> | 5.62E-01 | 4.92E-01 | -7.73E-02 | 1.78E-01 | 5.08E-01 | **6.91E-01** |
>
> Results show that 1) DesignX outperforms CMAES, SBS and the ablated baselines. 2) The single optimizer workflow found by SBS cannot defeat the workflows generated by Agent-1 with randomized Agent-2, validating the necessity of generating customized workflows for each problem.
>
> **[W5&Q5, Limitation Discussion]**
>
> We apologize for the absence of the discussion of limitations. Here we discuss the limitation of DesignX from the four aspects you suggested.
>
> - **Computational overhead / resource requirements:** We report the average solving time of DesignX, traditional BBO optimizer and MetaBBO basleines below: admit that compared to traditional BBO optimizers, DesignX has higher computational overhead due to additional agents. We report the averaged runtime for one test run on one problem of DesignX, BBO optimizers and MetaBBO optimizers below.
>
>
>     | DesignX | BBO | MetaBBO |
>     | --- | --- | --- |
>     | 5.5s | 2.6s | 3.6s |
>
>     Indeed, DesignX requires more computational overhead to solve a problem. However, the solving time remains at second-level. Considering that: a) DesignX achieves better optimization performance than baselines, and b) for human users who have limited knowledge of designing optimizers, the time and resources DesignX saves are actually implicit. This reveals certain advantage of our DesignX.
>
>     For hardware requirements, training DesignX requires two 32 cores CPUs and 488G RAM, while using DesinX to infer only requires a single CPU core.
>
> - **Conceptual complexity / tunability:** Compared with traditional optimizers, DesignX’s architecture and training paradigm are relatively complex. However, we have to clarify that this is the burden of us, DesignX’s developers, and is not in the development loop of DesignX’s users. With DesignX at hand, a user could solve optimization problems without cumbersome algorithm selection and tuning process. For the generalization aspect of DesignX, we admit more future works have to be explored such as a) improving its life-long learning/continual learning through robust fine-tuning strategy; b) in-depth study on relationship between problem space and generalization boundary.
> - **Evolutionary BBO:** Indeed, the primary focus of DesignX is currently restricted within EC methods. However, the methodology of constructing modular space, the universal interface design, and the corresponding training paradigm is generic. Take BO as an example, on the one hand, we could include algorithm modules in BO methods into Modular-EC thanks to our universal interface design. On the other hand, DesignX could also serve as an inspiration for BO developers to construct their BO version DesignX.
> - **Tuning of hand-crafted baselines:** We would clarify that we have tried two baseline settings: a) default settings, that is, following the settings in their original papers, and b) settings fine-tuned by SMAC3. In preliminary experiment we found that the fine-tuned performance is only comparable with the default settings. This is not surprising since the default choices typically have been set by the authors through careful sensitivity analysis. As a result, as stated in Appendix E.1, we use default settings for hand-crafted baselines.
>
> **[Paper Formatting Concerns]**
>
> We apologize for any confusion. We have addressed the typos you mentioned in the revised paper. The first matrix $\mathcal{W}\_{token} \in \mathbb{R}^{13 \times h}$ in 3.2.1 is a typo. It should be $\mathcal{W}\_{feature}$ indicating the problem feature embedder. The second $\mathcal{W}\_{token}\in\mathbb{R}^{16 \times h}$ is correct, indicating the tokenizer.
>
> Developing DesignX is a long long challenging journey, we really hope this study could appeal more researchers to explore possibility of foundation model for optimization. We hope the above responses could address your concerns and at last turning you back to acceptance side. Thanks for taking your time reading such long long responses. If you have any further questions, we promise timely address them in the next rebuttal phase.

---

> > ### Comment · Reviewer_WYcX · 2025-08-01
> >
> > I would like to thank the authors for their detailed response and following many of my suggestions and will carefully reconsider my evaluation at the end of the discussion phase.
> >
> >
> > [W1&W2&Q1&Q2, Positioning and Related Works Discussion]
> >
> > Thank you. Including these in your discussion of related work would address this concern.
> >
> > One note regarding rephrasing of the main contribution.
> > While Adding "deep learning" is an improvement, it is still very general and what exactly constitutes deep learning vague.
> > Some other earlier works would arguably classify, e.g.,
> > - AlphaD3M
> > - Auto-Sklearn 2.0
> >
> > Note that I am not challenging novelty of the work. However, am of the opinion that the core novelty lies in the fact that the tuning is "dynamic" (hyperparameters vary during the run) as in dynamic algorithm configuration. Updating the reformulation to clarify what constitutes 'tuning' would help address this concern, e.g., "To our knowledge, this is the first deep learning-based framework that jointly learns workflow generation and dynamic algorithm configuration in an end-to-end fashion". If you wish to stick closer to the original terminology, one could use "online tuning" or "parameter control", but I feel a direct reference to DAC would help positioning this work in the meta-algorithmics literature.
> >
> >
> > [W3&Q3, Better Statistical Presentation]
> >
> > Thank you for following my suggestion. These results definitely give me a better perspective on the relative performance of the methods compared.
> >
> > Based on these new results, it seems CMA-ES outperforms DesignX on a broad class of problems and performs competitive on average.
> >
> > Some follow up questions / concerns:
> > - How were the top 10 baselines chosen?
> > - Could you provide / include the rank-based comparison across all instances?
> > - Could you provide / include the rank-based comparison for other partitionings: conditioning, problem dimensionality and optimization budget?
> > - This result seems inconsistent with the claim that "The human-crafted BBO optimizers achieve progressive advancement through the expert-level designs proposed over the past decades" (CMA-ES is from 2001, and more recent methods rank considerably worse) - Could you comment on this?
> > - This result seems to cast doubt on the claim that "the DesignX-generated optimizers continuously surpass human-crafted optimizers by orders of magnitude" - will this weakened?
> > - Assuming CMA-ES is in the ModularEC space, why is it not generated for these "uni-modal" and or "adequately structured" problems?
> >
> > [W4&Q4, Ablation Baselines]
> >
> > Thank you for following my suggestion.
> >
> > Could you provide some additional details about the SBS baseline?
> >
> > Could you comment on
> > - SBS performing worse than CMA-ES?
> > - the significance of the small performance difference between SBS and DesignX w/o A2
> >
> > [W5&Q5, Limitation Discussion]
> >
> > Thank you. I think these limitations are acceptable and my concern would be addressed by including a discussion of these in the paper.
> >
> > Some follow up questions / comments regarding "computational overhead / resource requirements":
> >   - "the averaged runtime for one test run on one problem of DesignX, BBO optimizers and MetaBBO optimizers" - Could you clarify the measurement, did you select one specific problem or average across? Did you average across all BBO and MetaBBO methods. Could you provide runtimes for the top 10 baselines above? In particular, CMA-ES.
> >   - Note that BBO is often ran inside a loop (e.g., acquisition function optimization in BO), and doubling the runtime, even if a single optimization run only takes seconds, can be significant.
> >   - The "use vs design cost" argument assumes little reuse. While CMA-ES may have taken humans longer to design, this is a one time cost amortized by the fact that CMA-ES with default hyperparameters (your highly competitive baseline) is widely used. I fail to see how DesignX becoming the new default, while being slower than CMA-ES, would save resources in the long run.

---

> > > ### Author Response · Authors · 2025-08-02
> > > **Second Round Rebuttal with Reviewer #WYcX [Part I]**
> > >
> > > Dear reviewer #WYcX, appreciate for the timely feedback. We provide following responses and hope they could address your concerns well.
> > >
> > > ### [W1&W2&Q1&Q2, Positioning and Related Works Discussion]
> > >
> > > We agree, “deep learning” is indeed vague to some extends, we are happy to take your suggestion and rephrase the claim to “this is the first deep learning-based framework that jointly learns workflow generation and dynamic algorithm configuration in an end-to-end fashion”.
> > >
> > > ### [W3&Q3, Better Statistical Presentation]
> > >
> > > - The top-10 baselines were chosen according to their ranks on unimodel partition. We want to explain that for the other three listed partitions (multimodal, adequate and weak), we present these 10 baselines to make the comparison more intuitive. That is to say, the top-10 in these partitions vary with the top-10 in unimodal partition.
> > > - We provide the ranks of baselines across all instances below. The baselines are sorted by ranks, following the style of the rank-based statistical diagram. DesignX slightly outperforms CMAES and dominates other baselines. We also want the reviewer to consider that: ranks and objective value-based metrics should be comprehensively considered when BBO methods are compared with each other, because rank can not reflect accuracy of optimal an optimizer finds.
> > >
> > > | GLHF | CoDE | DEDQN | FIPSO | PSO | CLPSO | IPSO | GA | MMES | MadDE | GLPSO | DE | SHADE | GLEET | jDE21 | JADE | LMCMAES | NL-SHADE-LBC | SaDE | CMAES | DesignX |
> > > | --- | --- | --- | --- | --- | --- | --- | --- | --- | --- | --- | --- | --- | --- | --- | --- | --- | --- | --- | --- | --- |
> > > | 19.763 | 19.620 | 19.127 | 18.399 | 12.877 | 12.804 | 12.038 | 12.018 | 11.056 | 11.018 | 9.920 | 9.810 | 9.564 | 8.789 | 8.715 | 8.706 | 7.287 | 6.228 | 5.932 | 3.674 | 3.550 |
> > > - We report the full rank results of all baselines on modality, global structure, conditioning, dimensionality and optimization budget in tables below.
> > >
> > > **Unimodal**
> > >
> > > | GLHF | DEDQN | FIPSO | CoDE | CLPSO | PSO | GA | MadDE | IPSO | DE | GLPSO | GLEET | SHADE | jDE21 | JADE | MMES | NL-SHADE-LBC | LMCMAES | SaDE | DesignX | CMAES |
> > > | --- | --- | --- | --- | --- | --- | --- | --- | --- | --- | --- | --- | --- | --- | --- | --- | --- | --- | --- | --- | --- |
> > > | 19.274 | 19.261 | 18.801 | 18.793 | 13.572 | 13.270 | 12.657 | 11.756 | 11.743 | 10.513 | 9.901 | 9.901 | 9.117 | 8.783 | 8.743 | 8.541 | 7.332 | 5.518 | 5.392 | 4.276 | 3.850 |
> > >
> > > **Multimodal**
> > >
> > > | GLHF | CoDE | DEDQN | FIPSO | PSO | CLPSO | IPSO | GA | MMES | MadDE | GLPSO | DE | SHADE | GLEET | jDE21 | JADE | LMCMAES | SaDE | NL-SHADE-LBC | CMAES | DesignX |
> > > | --- | --- | --- | --- | --- | --- | --- | --- | --- | --- | --- | --- | --- | --- | --- | --- | --- | --- | --- | --- | --- |
> > > | 19.819 | 19.715 | 19.111 | 18.352 | 12.832 | 12.715 | 12.072 | 11.944 | 11.346 | 10.932 | 9.926 | 9.840 | 9.617 | 8.884 | 8.707 | 8.702 | 7.715 | 6.216 | 6.100 | 3.497 | 2.952 |
> > >
> > > **Adequate Global Structure**
> > >
> > > | GLHF | CoDE | DEDQN | FIPSO | PSO | CLPSO | GA | IPSO | MadDE | MMES | GLPSO | DE | SHADE | JADE | GLEET | jDE21 | LMCMAES | NL-SHADE-LBC | SaDE | DesignX | CMAES |
> > > | --- | --- | --- | --- | --- | --- | --- | --- | --- | --- | --- | --- | --- | --- | --- | --- | --- | --- | --- | --- | --- |
> > > | 19.811 | 19.667 | 19.107 | 18.418 | 12.908 | 12.846 | 12.309 | 12.203 | 11.197 | 10.626 | 10.160 | 9.761 | 9.351 | 8.959 | 8.861 | 8.835 | 7.184 | 6.351 | 6.130 | 3.544 | 2.771 |
> > >
> > > **Weak Global Structure**
> > >
> > > | GLHF | CoDE | DEDQN | FIPSO | MMES | PSO | CLPSO | IPSO | SHADE | DE | GA | MadDE | GLEET | LMCMAES | GLPSO | jDE21 | JADE | SaDE | NL-SHADE-LBC | CMAES | DesignX |
> > > | --- | --- | --- | --- | --- | --- | --- | --- | --- | --- | --- | --- | --- | --- | --- | --- | --- | --- | --- | --- | --- |
> > > | 19.471 | 19.333 | 19.246 | 18.280 | 13.696 | 12.693 | 12.546 | 11.023 | 10.880 | 10.830 | 10.226 | 9.920 | 9.777 | 9.353 | 8.466 | 7.980 | 7.156 | 6.136 | 5.473 | 4.870 | 3.640 |
> > >
> > > **Low Conditioning**
> > >
> > > | GLHF | DEDQN | CoDE | FIPSO | CLPSO | PSO | IPSO | GA | MadDE | DE | MMES | GLPSO | SHADE | GLEET | jDE21 | JADE | LMCMAES | NL-SHADE-LBC | SaDE | CMAES | DesignX |
> > > | --- | --- | --- | --- | --- | --- | --- | --- | --- | --- | --- | --- | --- | --- | --- | --- | --- | --- | --- | --- | --- |
> > > | 19.594 | 19.238 | 19.113 | 18.704 | 13.082 | 12.937 | 11.636 | 11.625 | 10.707 | 10.358 | 10.343 | 10.284 | 9.727 | 9.076 | 8.250 | 8.034 | 6.548 | 6.494 | 6.036 | 5.443 | 3.764 |
> > >
> > > **High Conditioning**
> > >
> > > | GLHF | CoDE | DEDQN | FIPSO | PSO | CLPSO | IPSO | GA | MMES | MadDE | GLPSO | DE | SHADE | GLEET | JADE | jDE21 | LMCMAES | NL-SHADE-LBC | SaDE | DesignX | CMAES |
> > > | --- | --- | --- | --- | --- | --- | --- | --- | --- | --- | --- | --- | --- | --- | --- | --- | --- | --- | --- | --- | --- |
> > > | 19.796 | 19.720 | 19.105 | 18.339 | 12.866 | 12.750 | 12.116 | 12.095 | 11.196 | 11.079 | 9.852 | 9.792 | 9.533 | 8.972 | 8.838 | 8.806 | 7.672 | 6.176 | 6.149 | 3.516 | 2.598 |

---

> > > ### Author Response · Authors · 2025-08-02
> > > **Second Round Rebuttal with Reviewer #WYcX [Part II]**
> > >
> > > **5D**
> > >
> > > | DEDQN | FIPSO | GLHF | MMES | CoDE | PSO | CLPSO | IPSO | GA | LMCMAES | GLEET | GLPSO | SHADE | jDE21 | DE | NL-SHADE-LBC | SaDE | JADE | MadDE | DesignX | CMAES |
> > > | --- | --- | --- | --- | --- | --- | --- | --- | --- | --- | --- | --- | --- | --- | --- | --- | --- | --- | --- | --- | --- |
> > > | 19.383 | 18.900 | 18.725 | 16.550 | 16.166 | 13.641 | 13.400 | 13.125 | 12.133 | 11.516 | 11.200 | 10.500 | 8.967 | 8.508 | 7.933 | 6.800 | 5.442 | 5.308 | 5.183 | 4.041 | 3.575 |
> > >
> > > **50D**
> > >
> > > | CoDE | GLHF | DEDQN | FIPSO | MadDE | JADE | DE | CLPSO | PSO | GA | IPSO | jDE21 | SHADE | MMES | GLPSO | GLEET | NL-SHADE-LBC | SaDE | LMCMAES | CMAES | DesignX |
> > > | --- | --- | --- | --- | --- | --- | --- | --- | --- | --- | --- | --- | --- | --- | --- | --- | --- | --- | --- | --- | --- |
> > > | 19.918 | 19.786 | 19.120 | 18.104 | 15.359 | 13.722 | 13.101 | 12.569 | 12.505 | 10.888 | 10.715 | 9.337 | 9.089 | 8.573 | 7.648 | 7.636 | 6.629 | 6.063 | 3.943 | 3.313 | 2.974 |
> > >
> > > **1E4 FEs**
> > >
> > > | CoDE | GLHF | DEDQN | FIPSO | GA | MadDE | GLPSO | MMES | SHADE | DE | PSO | IPSO | CLPSO | JADE | LMCMAES | jDE21 | SaDE | GLEET | NL-SHADE-LBC | CMAES | DesignX |
> > > | --- | --- | --- | --- | --- | --- | --- | --- | --- | --- | --- | --- | --- | --- | --- | --- | --- | --- | --- | --- | --- |
> > > | 20.330 | 19.361 | 19.258 | 17.553 | 12.313 | 11.714 | 11.401 | 11.401 | 11.393 | 11.312 | 11.223 | 10.973 | 10.232 | 8.758 | 8.719 | 8.339 | 8.152 | 6.866 | 4.821 | 3.616 | 3.259 |
> > >
> > > **5E4 FEs**
> > >
> > > | GLHF | CoDE | DEDQN | FIPSO | CLPSO | PSO | GA | IPSO | MMES | MadDE | GLEET | GLPSO | DE | jDE21 | SHADE | JADE | NL-SHADE-LBC | LMCMAES | SaDE | DesignX | CMAES |
> > > | --- | --- | --- | --- | --- | --- | --- | --- | --- | --- | --- | --- | --- | --- | --- | --- | --- | --- | --- | --- | --- |
> > > | 19.924 | 19.253 | 18.993 | 18.725 | 13.667 | 13.658 | 12.572 | 12.010 | 10.708 | 10.578 | 9.631 | 9.522 | 9.435 | 9.156 | 9.138 | 8.760 | 6.829 | 6.613 | 5.218 | 3.519 | 3.083 |
> > > - We would like first note that we are not the first MetaBBO work that find CMA-ES outperforms more recent optimizers. We locate a possible reason behind: with intuitive workflow and mathematically guaranteed convergency, CMA-ES could provide robust optimization performance on a great portion of the test set. In contrast, more recent optimizers hold relatively customized design to overfit specific benchmark (e.g., winner of IEEE CEC 2021 competition, MadDE).
> > > - We think the rank results above would not weaken our claim, since ranks only reflect relative performance across optimizers and heavily rely on the selected baselines. We have provided the overall and average absolute optimization performance results (based on objective values) in the paper and online, where DesignX surpasses baseline optimizers including CMA-ES by orders of magnitude on a certain portion of tested problems (especially those more complex ones) during its training. We will revise the claim to “on a certain portion of tested problems” for rigorous purpose.
> > > - This is a quite in-depth question. Let us explain for you: 1) for some unimodal and adequately structured problems, we have double checked the workflows DesignX generates for them, of which a certain portion (31%) are CMA-ES like workflows; 2) for those DesignX generating other style workflows, we think the major reason roots from how we profile an optimization problem. In DesignX, we use well-known ELA features which are low-level value-based features. However, the unimodal/multimodal properties are high-level features that may not be perfectly classified by such ELA features, which subsequently triggers our DesignX to output other optimizers; 3) RL’s sampling inefficiency is another possible reason, the algorithm space expressed by our Modular-EC is huge, with the limited learning budget, learning variance might happens.
> > >
> > > ### [W4&Q4, Ablation Baselines]
> > >
> > > - SBS: The SBS baseline is closely following your suggestion, we ‘retrain DesignX without instance/observation features, causing it to effectively learn a single algorithm template and static setting’. We observe this baseline performs worse is because if we limit the freedom of RL sampling in Agent-1, the overall training collapses quickly and converges to unsatisfactory sampling trajectory.
> > > - DesignX w/o A2: We kindly remind the reviewer that the performance metrics have been normalized by a normalization term, which is the optimal found by random search (Appendix E.2). In such case, suppose for a problem, optimal found by random search is 10, while SBS’s optimal is 0.001 and  DesignX w/o A2’s optimal is 0.00001, then their normalzied performance is (1-0.001/10) and (1-0.00001/10), which is not significant at all, however, they actually differ by orders of magnitude. SBS underperforms  DesignX w/o A2 also reveals that choosing a correct workflow is significantly important for solving optimization problems.

---

> > > ### Author Response · Authors · 2025-08-02
> > > **Second Round Rebuttal with Reviewer #WYcX [Part III]**
> > >
> > > ### [W5&Q5, Limitation Discussion]
> > >
> > > - We average all problem instances, all BBO baselines and all independent runs for BBO’s 2.6s result. We average all problem instances, all MetaBBO baselines and all independent runs for MetaBBO’s 3.6s result. We average all problem instances and all independent runs for DesignX’s 5.5s result. We provide average run time of all baselines across all problem instances and independent runs in the following table. In particular, for CMA-ES, we use ‘pycma’ repo in Github, and its 5.0s result reveals that DesignX shows no significant downside in efficiency.
> > >
> > >
> > >     | GA | PSO | DE | CMAES | FIPSO | SaDE | CLPSO | JADE | CoDE | IPSO | SHADE | LMCMAES | GLPSO | MadDE | jDE21 | MMES | NL-SHADE-LBC | DEDQN | GLHF | GLEET | DesignX |
> > >     | --- | --- | --- | --- | --- | --- | --- | --- | --- | --- | --- | --- | --- | --- | --- | --- | --- | --- | --- | --- | --- |
> > >     | 1.5s | 1.2s | 1.4s | 5.0s | 1.3s | 1.6s | 1.4s | 1.6s | 1.6s | 1.3s | 1.6s | 3.9s | 2.1s | 4.9s | 4.9s | 3.9s | 5.1s | 2.4s | 5.9s | 2.9s | 5.5s |
> > > - We agree that ‘BBO is often ran inside a loop (e.g., acquisition function optimization in BO)’, which we will underline it as a limitation in the revised paper.
> > > - We would like to clarify that we did not claim that DesignX becomes the new default sota optimizer. Instead, we think DesignX is the first step to explore the possibility of whether foundational optimization model could relieve the the hand-crafted design process. Currently, DesignX can already be regarded as an alternative alongside sota optimizers such as CMA-ES when a user has little expertise in optimization (considering that our DesignX shows potential in more complex scenarios, and with on average similar running time with CMA-ES, 5.5s v.s. 5.0s). In the future, with the efforts we continuously make to address existing technique bottlenecks, we believe foundational optimization model will finally achieve “fully automated optimization algorithm design”.
> > >
> > >
> > > Again a long long response, thanks for your patience and time! Should there be any further questions, let us know.

---

> ### Comment · Reviewer_WYcX · 2025-08-04
>
> I would like to thank the authors for their extensive reply.
>
> [W1&W2&Q1&Q2, Positioning and Related Works Discussion]
> That change would close this concern.
>
> [W3&Q3, Better Statistical Presentation]
>
> Thank you for the full set of ranks.
>
> Based on your results, CMA-ES and DesignX are close competitors, but DesignX seems more sample-efficient / to perform better on complicated functions - which is both desirable and interesting.
>
> > We think the rank results above would not weaken our claim
> > We have provided the overall and average absolute optimization performance results (based on objective values) in the paper and online
>
> My main concern, at this point, is that I have some trouble reproducing similar results using the data you have originally provided in the excel sheet (anonymous repo):
>
> I found that
> - in a pairwise comparison, CMA-ES outperforms DesignX in 2780 of the 3200 test cases.
> - average rank (across all 21 methods, averaged across all 3200 functions) is CMA-ES: 3.52 vs DesignX: 7.89. In fact, 4 methods have better average ranks than DesignX: CMA-ES, SADE, LMCMAES, NL_SHADE_LBC.
> - based on average performance (objective) across all instances, CMA-ES ranks 1st and DesignX only 10th out of 21 methods. Note that this metric is extremely susceptible to outliers, but still it is a metric used by the authors.
>
> Is this consistent with your observations?
>
> Note:  I found the excel sheet somewhat difficult to parse as "value (error)" is provided as text for each method / function pair. I extracted the values and based my calculations on these. My apologies if something went wrong.
>
> [W4&Q4, Ablation Baselines]
>
> > We kindly remind the reviewer that the performance metrics have been normalized by a normalization term, which is the optimal found by random search (Appendix E.2). In such case, suppose for a problem, optimal found by random search is 10, while SBS’s optimal is 0.001 and DesignX w/o A2’s optimal is 0.00001, then their normalzied performance is (1-0.001/10) and (1-0.00001/10), which is not significant at all, however, they actually differ by orders of magnitude.
>
> I agree that such differences often matter and that the metric chosen by the authors does not properly reflect them (sensitivity to outliers) - which was my main concern in the evaluation and presentation. Also here, average rank would offer a different (better?) perspective. Could you provide ranks of the ablated versions? (this would also allow non-parametric / rank-based statistical tests)
>
> [W5&Q5, Limitation Discussion]
>
> Thank you for the more detailed runtime data.
>
> > We would like to clarify that we did not claim that DesignX becomes the new default sota optimizer. Instead, we think DesignX is the first step to explore the possibility of whether foundational optimization model could relieve the the hand-crafted design process. Currently, DesignX can already be regarded as an alternative alongside sota optimizers such as CMA-ES when a user has little expertise in optimization (considering that our DesignX shows potential in more complex scenarios, and with on average similar running time with CMA-ES, 5.5s v.s. 5.0s). In the future, with the efforts we continuously make to address existing technique bottlenecks, we believe foundational optimization model will finally achieve “fully automated optimization algorithm design”.
>
> I mostly agree with this argument and view this as an acceptable limitation. In my opinion, the original manuscript was not sufficiently transparent about DesignX just being "the first step". If the authors include such argument in the final version, this concern would be addressed.

---

> > ### Author Response · Authors · 2025-08-04
> > **Third Round Rebuttal with Reviewer #WYcX**
> >
> > We are happy certain concerns of your remaining concerns have been adressed by our previous responses (e.g., W1, Q1, W2, Q2, W5, Q5).  We appreciate your careful checking on the results we provided in anonymous repo, however, there are several clarifications we have to make to clear your remaining concerns on the results clarity.
> >
> > [W3 & Q3, reproducibility] We believe there is certain misunderstanding on our provided results and our observations.
> >
> > - The excel file we provide is the absolute average objective values (less is better) of a given baseline on a problem instance “across 51 independent runs”, the reason we provided this table is to facilitate later checking of reviewers. Hence, this is not the “metadata” we should use to compute the rank statistics you suggested. the rank statstics we compute is based on a more larger metadata, where for each baseline, we have runs (51) x instances (3200) columns for each baseline, and the rank results we provided above is obtained by examining the significance in such case, not the averaged 3200 columns in the excel.
> > - “based on average performance (objective) across all instances, CMA-ES ranks 1st and DesignX only 10th out of 21 methods. ”, this is not consistent with our results. Let us explain for you, again, the excel is the averaged absolute objective values of a baseline on a problem instances. For different problem instances, the objective values hold different scales, hence they can not be simply averaged further to obtain any rank information.  Instead, they are used to compute the normalized averaged performance across diverse problem instances as what we did in the last row of Table 1 of our paper.
> > - We hope the above explanation could address the results inconsistency between us and you. For the rank reproducibility, we promise to upload the runs (51) x instances (3200) tables for each baselines once the revision window opens. We appreciate the reviewer for giving us the rank presentation suggestion, we will definitely include all results about ranks to the revise paper (at least Appendix). Also thanks for reminding us to upload the per run results  (runs (51) x instances (3200) tables), we respectfully request you to check the uploaded results when we are allowed to make revisions.
> >
> > [W4 & Q4, ablation] We would like clarify that we fully agree with the reviwer the necessity of rank-based indicators, and we will remind readers in the revised paper to check the rank-based results to get different perspectives. Below, following your suggestion, we use the 51 runs x (3200 synthetic and 1271 realistic) problem instances columns of all ablated baselines to obtain the rank-based results among them. Note that we are not sure if the reviewer want to check the rank among “all” (baselines + ablated baselines) or only among the ablated baselines. So we show the results only among the 6 ablated baselines. Note that we will also upload a new table on all results in all tested runs as an excel file once we are allowed to make revisions.
> >
> > | w/o A1+A2 | w/o A1 | SBS | w/o A2 | CMAES | DesignX |
> > | --- | --- | --- | --- | --- | --- |
> > | 4.457 | 4.263 | 4.154 | 3.901 | 2.207 | 2.018 |
> >
> > [W5&Q5, Limitation Discussion] We will include “the first step” to clarify the positioning of current DesignX, thanks for your suggestion.
> >
> > We hope these responses could address your concerns. We will make all promised revisions when we could. We sincerely look forward to your positive feedback!

---

> > > ### Comment · Reviewer_WYcX · 2025-08-05
> > >
> > > Thank you once again for the clarifications and continued discussion.
> > >
> > >
> > > [W3 & Q3, reproducibility]
> > >
> > > > The excel file we provide is the absolute average objective values (less is better) of a given baseline on a problem instance “across 51 independent runs”
> > >
> > > Ok, so the values I used are averages per instance (across 51 seeds) before normalization, that indeed partially explains the discrepancies.
> > >
> > > > For different problem instances, the objective values hold different scales, hence they can not be simply averaged further to obtain any rank information.
> > >
> > > I fully agree that since values are not normalized, the averages across instances are effectively meaningless.
> > >
> > > However, **could you still briefly comment on / confirm my other observations**:
> > > - in a pairwise comparison, CMA-ES outperforms DesignX in 2780 of the 3200 test cases.
> > > - average rank (across all 21 methods, averaged across all 3200 functions) is CMA-ES: 3.52 vs DesignX: 7.89. In fact, 4 methods have better average ranks than DesignX: CMA-ES, SADE, LMCMAES, NL_SHADE_LBC.
> > >
> > > (FYI: This matters for my final re-evaluation, and I would not want to base this off my possibly incorrect interpretation of your results)
> > >
> > >
> > > [W4 & Q4, ablation]
> > >
> > > Thank you for providing the rank-based results. This addresses this concern.

---

> ### Author Response · Authors · 2025-08-06
> **Fourth Round Rebuttal with Reviewer #WYcX**
>
> We are happy that after three rounds of discussion, we have addressed most of the reviewer’s concerns. For the last concerns about the results evaluation/interpretation, we would like to clarify following aspects:
>
> - We first confirm consistent observations with you: 1) “in a pairwise comparison, CMA-ES outperforms DesignX in 2780 of the 3200 test cases”; 2) “average rank (across all 21 methods, averaged across all 3200 functions) is CMA-ES: 3.52 vs DesignX: 7.89”. These results are obtained by comparing averaged per-problem-instance objective values (without normalization), which to some extends, are also reasonable metrics could be used for evaluating BBO optimizers.
> - Although in the pairwise comparison, CMA-ES outperforms DesignX on 2780 test cases, this is just one perspective from rank of average objective, however, as we explained in previous rebuttal round, a more reasonable pairwise comparison should be conducted before average. In that case, on 51 independent runs of 3200 problem instances, DesignX outperforms CMA-ES on 68% of 51x3200 columns. The reason that results in such inconsistency is somehow tricky. While DesignX has 68% possibility to outperform CMA-ES with small performance gap, it might has 32% possibility to underperform CMA-ES with larger performance gap.
> - We believe the above reason could also explain  “average rank (across all 21 methods, averaged across all 3200 functions) is CMA-ES: 3.52 vs DesignX: 7.89”.
>
> As we discuss with the reviewer in these four rounds of discussion, we have a clearer and more objective perspective on how to evaluate/measure BBO optimizers. We believe this discussion is very important to our community, since with different metrics, the comparison results of baselines might go to a totally opposite way. This discussion must be added into our paper to appeal more and more participants rethink their evaluation protocols. Thanks for the patience you spend on our paper, your comments/suggestions indeed help us improve our paper, and more importantly, ourselves. We would like to reflect your valuable contribution for our paper to area chair, you deserve it!

---

> > ### Comment · Reviewer_WYcX · 2025-08-07
> >
> > Thank you for responding to my final request.
> >
> > FYI: One explanation for these observations could be that (relatively) poor performance on one or a few seeds/runs - pull down these averages for DesignX.
> >
> > Reference my original review:
> >
> > > I have serious concerns that in my view render the manuscript as-is unsuitable for publishing. That being said, if all these concerns are fully addressed, I am open to considerably increase my evaluation score and “accept” the paper.
> >
> > The authors' prompt and elaborate responses have addressed most of my original concerns and assuming the promised changes are implemented, I do not see further ground for rejection. I will improve my score from reject (2) to borderline accept (4). For a better score, I am missing a clear / consistent improvement w.r.t. e.g. CMA-ES which, given the complexity of the approach, I believe one could reasonably expect.

---

> > > ### Author Response · Authors · 2025-08-07
> > > **Thank Reviewer #WYcX**
> > >
> > > We sincerely appreciate reviewer #WYcX for the insightful and comprehensive review. We also enjoy the in-depth discussion with the reviewer on some aspects of our DesignX. These valuable discussions surely contribute to our paper and community, we will include them in the revised paper and keep the limitations of DesignX in mind to address in our future work. Thank you for your acceptance of our paper.

---

### Official Review · Reviewer_CRYE · 2025-07-02

**Clarity:** 4
**Significance:** 4
**Originality:** 4
**Rating:** 5
**Confidence:** 3

**Summary:**

This paper presents DesignX, an automated algorithm design framework that can automatically generate optimization algorithms to black-box optimization (BBO) algorithms. DesignX employs two agents, namely an algorithm structure generation agent (Agent 1) and a parameter tuning agent (Agent 2). Agent1 is responsible for taking the relevant features of the BBO problem and generating a program structure, while agent 2 is responsible for tuning the hyper parameters. The optimization of the two agents is formulated as a dual-agent MDP problem and trained cooperatively.

The agents are trained on an augmented BBO problem dataset and evaluated on both in-distribution test problems and realistic out-of-distribution problems. Results show that algorithms discovered by DesignX show superior or competitive performance compared to the baseline optimization algorithms, some of which are designed by expert humans. Ablation studies are conducted to test the effectiveness of both agents and additional analysis is presented to demonstrate the design principle of the automatically generated algorithms.

**Questions:**

1. How expandable is DesignX? If more advanced expert-designed BBO algorithms are available, would it be possible to incorporate them into the pipeline?
2. If the user is interested in a particular type of (or even just one) BBO problem, would it be possible to train or even fine tune DesignX for their usage?
3. The training objectives of DesignX include the “optimal objective value” (Line 218). Is this value computed and updated along the training or is it pre-computed? If it is pre-computed, how to ensure its optimality if a solution to the targeting BBO problem is not yet available?
4. How sensitive is DesignX to the dimensionality of the BBO problems. In the training and testing set, the dimensions of the BBO problems are all under 50. What if the dimension can exceed 1,000 or even 10,000?
5. The training of DesignX seems to require function evaluation of the BBO problems because the reward is dependent on the objective value. If so, how much slow down will DesignX incur if the BBO problem of interest requires expensive evaluation?

**Ethical Concerns:**

["NO or VERY MINOR ethics concerns only"]

**Final Justification:**

The authors' rebuttal addressed all of my concerns and questions on this paper. This paper has reasonable technical contents and they also have a reasonable plan for the future direction. I am happy to recommend an acceptance for this work.

**Limitations:**

Yes

**Quality:**

4

**Strengths And Weaknesses:**

## Strengths:
1. The idea of searching for novel algorithms for BBO problems is interesting. Given the large space of BBO problems, it would be exciting to have an automated framework that can generate algorithms for them. Even if the generated algorithms might be suboptimal, it can provide the human with insights in how to design better algorithms.
2. The presentation of the paper is clear and easy to follow. I like how Fig2 gives an intuitive and detailed overview of the framework.
3. The experiment section gives comprehensive evaluation of the framework. The design principles revealed from DesignX that initialization is not important (according to the algorithm) is interesting.

## Weakness:
1. In addition to the training time of DesignX, I suggest presenting the compute resource required to train DesignX, so that the users have a better idea of the resource needed.
2. Minor: DE seems to refer to Differentiable Evolution, but it is never mentioned in the main paper. I suggest mentioning the full name when it is referred to for the first time.

---

> ### Author Rebuttal · Authors · 2025-07-30
>
> We appreciate the reviewer for your positive and valuable feedback. For the remaining concerns, we address them as below.
>
> **[W1, Computational Resource Requirement]**
>
> Thank you for your valuable suggestion, we would clarify that the computational resources used for DesignX's training are reported in Appendix E.1: we train DesignX on two Intel(R) Xeon(R) 6458Q CPUs with 488G RAM.
>
> **[W2, Abbreviation Issue]**
>
> We apologize for the overuse of abbreviations. We will follow your suggestion and include the full names of all abbreviations upon their first use in the revised paper.
>
> **[Q1&Q2, Extendibility and Transferring]**
>
> Modular-EC is an extensible framework where modules are organized within a hierarchical inheritance architecture and universal interfaces. If more advanced expert-designed BBO algorithms are available, users can decompose algorithms into modules, integrate these modules into Modular-EC by inheriting its base classes and implementing modules using universal interfaces. This approach also works for integrating modules tailored to specific types of BBO problems, such as Bayesian optimization modules for solving expensive problems. With these new modules, similar to ConfigX [1], users can conduct life-long learning to adapt the DesignX model to new modules and solve problems of interest with advanced expert-designed BBO modules.
>
> [1] Guo, Hongshu, et al. "Configx: Modular configuration for evolutionary algorithms via multitask reinforcement learning." *Proceedings of the AAAI Conference on Artificial Intelligence*. Vol. 39. No. 25. 2025.
>
> **[Q3, Optimal Objective Value]**
>
> The training problems are synthetic problems from well-known optimization benchmarks, including the CoCo BBOB and CEC benchmarks. These problems have known optimal objective values. In this paper, these values are 0. If the optimal objective value for the target BBO problem is not yet available (e.g., for realistic benchmark problems), we use the best objective value found by all baselines across all runs as the "optimal objective value".
>
> **[Q4, Dimensionality Sensitivity]**
>
> In this paper, Modular-EC integrates BBO modules tailored to low-dimensional optimization. For large-scale optimization where dimensionality can exceed 1,000 or even 10,000, DesignX with the current Modular-EC cannot handle these problems. As mentioned in the response to Q1 & Q2, Modular-EC is an extensible framework. We can integrate large-scale optimization modules and conduct transfer learning to enable DesignX to solve high-dimensional problems, which is a future work.
>
> **[Q5, Efficiency on Expensive Problems]**
>
> BBO and MetaBBO optimizers require objective values during optimization; therefore, expensive problems reduce their efficiency. Since DesignX’s reward is calculated using objective values obtained directly from the low-level optimizer without extra evaluations, its incurred slowdown is equivalent to that of both BBO optimizers and other MetaBBO optimizers. To address expensive evaluations, surrogate-based methods have been extensively studied. DesignX could integrate these optimization modules to enhance efficiency for expensive problems.
>
> We hope the above responses could address your concerns and anticipate your timely feedback.

---

### Official Review · Reviewer_kkQ7 · 2025-07-03

**Clarity:** 3
**Significance:** 3
**Originality:** 4
**Rating:** 5
**Confidence:** 4

**Summary:**

This paper presents designX, which is an automated algorithm design benchmark that can learn optimizer for BBO problem efficiently. The authors created a comprehensive algorithmic space that incorporates hundreds of potential related tasks / questions. Then,  a two-agent reinforcement learning approach is proposed to learn from the generated algorithmic space. The proposed learning agents achieve effective performance improvement on the designated benchmarks.

**Questions:**

- What is the effect of using non-transformer architectures? Any rationale here?

- Maybe I am wrong, but how is the dual-agent system being able to learn on two CPUs system? Shouldn't it be a mixture of GPU + CPU since the agents are GPT-2 like transformers. The author should clarify on this point.

**Ethical Concerns:**

["NO or VERY MINOR ethics concerns only"]

**Final Justification:**

The authors provided a comprehensive rebuttal to my questions. After the discussion, the authors promised to provide revision to the comments I made. These revisions would presumably solve most of my concerns. Therefore, I recommend to accept this paper.

**Limitations:**

Author didn't mention limitation.
Quoted"The limitations of the work are not discussed due to page limitation."

**Quality:**

4

**Strengths And Weaknesses:**

Strength
+ Idea looks pretty interesting, consideration of designing dual-agent looks logical and reasonable.
+ the efforts on designing the algorithmic space should be helpful for the community.
+ results seem to be effective.
+ some of the observation of what DesignX was learned is interesting.
+ good efforts on open sourcing.

Weakness
- the design space can still be further extended. In fact, I might interpret the maximum # of action (12?) and # of operation (16?) through the code, but it seems like the author doesn't claim much on the reason to pick certain action or operation pools and to clarify the coverage of their choice is good. If the author mentioned in supplementary materials, I would encourage them to clearly point it out, if not, I think this part needs to be further justified. The authors should also discuss the potential extendibility of the proposed work, especially when the targeted problem is much more complex that the existing operations might not  be enough to serve as a support design choice pool.
- parts of the illustration of the agent design is somewhat hard to understand. For example, in Figure 2, are the GPT-2 blocks shared by both agents? The same color used might cause such confusion. From the code it seems like they are separate models, but I wonder is the design of these 2 agents share any similarities / differences?
- The claim said designing such optimizers requires up to month sounds like an overclaim, especially considering all experiments were conducted on two CPUs. The authors can clarify on this.
- didn't mention limitation

---

> ### Author Rebuttal · Authors · 2025-07-30
>
> We sincerely appreciate the reviewer for your insightful comments and valuable suggestions. For your concerns, we provide following point-to-point responses to address them.
>
> **[W1, Design Space Extendibility and Coverage]**
>
> - Coverage of Modular-EC: we would like to clarify that the operator pool choice and action length choice have provided a wide coverage over existing single-objective optimizers. For operator pool, the 116 submodules are carefully collected by us from decades of literatures and representative BBO optimizers. We provide two DE examples below to showcase Modular-EC could express either simple structure or very complex modern optimizers:
>     1. a simple 1997 DE/rand/1/bin combines:
>         1. Uniform initialization module
>         2. rand/1 mutation module
>         3. binomial crossover module
>         4. Clip boundary control module
>         5. DE-like selection module
>     2. a very advanced and complex 2022 NL-SHADE-LBC optimizer uses:
>         1. Uniform initialization module
>         2. current-to-pbest/1 with archive mutation module
>         3. binomial crossover module
>         4. Halving boundary control module
>         5. DE-like selection module
>         6. Non-linear population reduction module
>
>     For action length choice (12), we believe it is enough for existing or any potential operators, e.g., even the latest DE algorithms such as  NL-SHADE-LBC (winner of IEEE CEC Congress) holding 3-4 parameters per module.
>
> - Extendability for Outlier Problems: On the one hand, thanks to our proposed universal interfaces and inherentance architecture in Modular-EC, we believe extending it to embrace other novel algorithmic modules is quite easy to achieve. On the other hand, the 16-bit binary module ID spans a huge module space, many of which are still reserved for future module extension. We agree with the reviewer that extending (or more specifically fine-tuning) DesignX for complex optimization domains still require efforts such as novel fine-tuning strategy. For such point, we cite the paper ConfigX [1], where life-long learning ability of similar modular system has been validated. We will regard this as an important future work to enhance the scalability of DesignX.
>
> **[W2, Agent Model Comparison]**
>
> Agent-1 and Agent-2 use separate GPT-2 models without shared parameters. We will clarify this in the revised paper.
>
> 1. Similarity: the internal GPT-2 blocks share same architecture.
> 2. Difference: First, the input and output of the two agents differ with each other: for Agent-1, input is the problem feature, output is the softmax distribution of submodules; for Agent-2 input is the concat of submodule-id and optimization status, output is the sample distribution of hyper-parmeter values. Second, the feedforward logics are different: for Agent-1, the GPT-2 auto-regressively output submodules; for Agent-2, it is a self-attention among submodules in the generated workflow.
>
> **[W3, Designing Time]**
>
> We would like to clarify that when we claim “designing such optimizers requires up to month”, we mean that for humans, designing an optimizer and fine-tune its hyper-parameters to achieve maximal perforamence gain on the target problem might consume days to months. Humans may select a BBO optimizer suitable for the target problem according to their expert-knowledge on optimization, design optimizer’s workfow, fine-tune the hyper-parameters and evaluate the optimizer on the target problems. Such process would repeat over and over again until the optimizer achieves desired performance on the target problem, which is really time-consuming.
>
> In contrast, our trained DesignX only consumes seconds to infer an algorithm and optimize the given problem. If we understood your question wrongly, please give us timely feedback so we can discuss with you more on this issue.
>
> **[W4&Limitations, Discussion on Limitation]**
>
> We appreciate your careful reading and constructive suggestion. We will replenish a limitation discussion section into the appendix of the revised paper to enhance this paper’s scientific integrity.
>
> > While Designx has been validated as an effective and novel framework to learn automated algorithm design policy, it still hold two limitations and anticipates further efforts in the future to address them.
> >
>
> > 1. In this paper, the primary optimization domain is restricted within single-objective optimization. However, optimization domains such as multi-objective optimization requires more complex algorithm modules and corresponding training problems, which challenges the extensibility of DesignX. Thanks to the uniform interfaces we have developed in Modular-EC system, complex modules in such optimization domain can be easily integrated, and life-long learning/continual learning techniques might be a promising way to make DesignX embrace diverse optimization domains.
> >
>
> > 2. The training efficiency of DesignX is not very ideal. It still takes us 6 days to train DesignX on high-performance PC. Further improvement such as Multi-card parallel mechanism, proximal training objective construction, architecture simplefication/distillation could be regarded as important future works.
> >
>
> We hope the above discussion could address your concerns. We will refine such discussion into the revised paper.
>
> **[Q1, Rationale of Transformer Architecture]**
>
> Let us explain the rationale behind our Transformer choice for you. First, we actually have conducted prelimary experiments on other neural network architectures, to be specific, MLP and LSTM architectures. Four experiments were conducted:
>
> - Agent-1 (MLP), Agent-2 (MLP): this experiment fails since MLP can not generate algorithm workflow in an auto-regressive fashion.
> - Agent-1 (MLP), Agent-2 (LSTM): same as above, fails.
> - Agent-1 (LSTM), Agent-2 (MLP): we observe two issues in this experiment, during training of Agent-1, the LSTM encounters gradient explosion due to the MDP of Agent-1 is an reward-delayed one, hence the policy gradient might be too much for LSTM. The parameter control performance of MLP-based Agent-2 is far worse due to MLP can not capture the topology dependency within an complete algorithm workflow.
> - Agent-1 (LSTM), Agent-2 (LSTM): Agent-1 still encounters gradient explosion issue while LSTM-based Agent-2 performs better than MLP-based one. However, it still underperforms DesignX.
>
> Second, these preliminary results coherent with a recent MetaBBO work ConfigX [1], where the ablation results also reveal that Transformer architecture achieves superior modelling capabability than MLP and LSTM.
>
> **[Q2, Training Hardware]**
>
> We indeed train our Design on 2 32-core CPUs with 488G RAM. Our primary consideration regarding GPU deployment for agents involves the trade-off between GPU acceleration of network operations and CPU-GPU transfer overhead. The total number of parameters of DesignX’s Agent-1&2 are 100k. For such relatively small models, GPU acceleration cannot offset the overhead introduced by excessive CPU-GPU transfers. Therefore we do not use CPU-GPU mixing strategy. More importantly, during the testing, for a single testing problem, 1 CPU core is enough to infer the algorithm workflow and do hyper-parameter control in the subsequent optimization episode. However, for future work involving larger training sets or models with more parameters, GPU deployment may be efficiency.
>
> We hope the above responses could address your concerns and anticipate your timely feedback.
>
> [1] Guo, Hongshu, et al. "Configx: Modular configuration for evolutionary algorithms via multitask reinforcement learning." *Proceedings of the AAAI Conference on Artificial Intelligence*. Vol. 39. No. 25. 2025.

---

> > ### Comment · Reviewer_kkQ7 · 2025-08-01
> > **Follow up on the rebuttal**
> >
> > Thanks the authors for the comprehensive response. I will follow up with authors points by points. Note that I may focus on several questions first and then turn to other questions. That being said, my question may not come in numerical order. I will make a final revision after the discussion ends.
> >
> > [W3, design time] Thanks the author for the detailed explanation and their perspective. However, this does sound as an overclaim and put unfair constraint on the human side. When the comparison includes human ability, say a comparative study, the evaluation metric for human ability needs to be clearly defined and measured. For example, the expertise, the task design, the human participant recruiting strategy, the inclusion / exclusion standard, the evaluation protocol, needs to be clearly defined. I understand the authors may infer the time based on their experience to make such a claim, but such claim is less rigorous, given the authors provided no detailed comparative study.
> >
> > One example could be, say if a human participant is given a runnable algorithmic script for a basic optimizer, how long does it really take to reach a reasonable performance on such tasks? I feel depending on the setting, the time may vary from hours to months (if the setting is really really hard), but the evidence is not shown in this paper. I strongly suggest the reviewer reconsider their claim and revise.
> >
> > [Q1&Q2] Thanks the author for their response. So first, correct me if I am wrong, but if the experiments were conducted on 2 CPU-major hardware, this probably indicates that the 'GPT-2' the authors used in this work, is probably not the identical 'GPT-2' architecture with same layer / parameter configuration (authors mentioned it is a small model with 100K learnable parameter). I would doubt the feasibility to train a standard 'GPT-2' model only requiring CPUs. The authors may use the atomic architecture design but with a shrunk configuration. The authors should disclose the detailed model architecture, and consider revise to avoid potential confusion.
> >
> > Given the interpretation I had above, I am more curious about the reason why training DesignX counts for 6 days on CPU machine. Can the authors provide a break down for the computation overhead in one atomic training step? This would be insightful to understand the overall landscape of the proposed work.
> >
> > For Q1, I like the authors' response on disclosing their design choice. Are the selected architecture having similar parameters configuration? Can the authors provide a bit more information on this?
> >
> > I would also like the authors to consider provide a discussion (at least here), about the thoughts on claiming 'design times' vs. the actual computation overhead. In the recent reality, the computation infrastructure design tends to shift to a more GPU-oriented nature especially for machine learning based discovery applications. In this work, the required computation hardware doesn't sound like a hard piece to achieve, I wonder how authors position the wider applicability of their method and foresee the future of their work.

---

> > > ### Author Response · Authors · 2025-08-02
> > > **Second Round Rebuttal with Reviewer #kkQ7 [Part I]**
> > >
> > > Thanks for your timely feedback. We appreciate the reviewer for providing us a second oppotunity to further address your remaining concerns and polish our paper accordingly.
> > >
> > > ### [W3, design time]
> > >
> > > We think the reviewer provides us a very very important insight on how to subjectively compare AI with human for algorithm design domain, which we have to mention, is barely mentioned in existing literature.  We agree that rigorous human-in-loop comparison as you suggested is necessary, which outlines an important future work of DesignX: including diverse human developers with diverse designing behaviour to quantify the design time.
> > >
> > > However, we would like also clarify that we indeed have some preliminary comparison tests to back up our claim. In the test, we assume a human participant has an exsiting optimizer (traditional baselines in our experiments) at hand. For all tested problems, we call the interface of SMAC3 (BO-based hpo toolbox) to configure the optimizer for reasonable performance improvement. On average, it require 2-3 hours to configure the optimizer per problem instance, while the performance improvement is still limited (statistically underperforms DesignX). We believe the above scenario at least resembles the real-world human-in-loop design process. Another test is we reproduced a scenario where the user might has no optimization prior, means that he might randomly selects a baseline and directly solve the problem. In such case, we approximately estimate the user need 1-2 minutes to ask internet (or GPT) to obtain a list of algorithms and their characteristic descriptions, then he may also need 5-10 minutes (at least..) to demtermine one to use. According to our main comparison results and ablation study (DesignX w/o A1 resembles randomly algorithm selection), we believe our claim is supported to some extends.
> > >
> > > Nevertheless, we will revise out claim to not mention the hours and days, instead we will claim that DesignX relieves the labor-intensive human design loop through in an end-to-end fashion. We will also add the above discussion into the revised paper for future readers.
> > >
> > > ### [Q1, specific NN configuration]
> > >
> > > We understand your concerns and are happy to clarify these details for you:
> > >
> > > - GPT-2: You are right, while our GPT-2’s architecture is exactly the OpenAI’s official implementation,  the configuration parameters are adjusted by us. The configurations of OpenAI is 12 layers with 12 heads and 768 hidden-dim each layer.  In our case, we use 3 layers with 4 heads and 64 hidden-dim each layer for out agents. Training original GPT-2 in 2 CPUs is impractical, which is the major reason we customized our own configurations. (Also due to our limited computational resources, we hope the reviewer could consider such aspect..)
> > > - Considering the MLP/LSTM preliminary study, we ensured similar parameter scales by using a 5-layer MLP ([64 X 128 X 192 X 128 X 64]) with relu activation and a 2-layer LSTM (64 input-dim, 64 hidden-dim).

---

> > > ### Author Response · Authors · 2025-08-02
> > > **Second Round Rebuttal with Reviewer #kkQ7 [Part II]**
> > >
> > > ### [Q2, 6 days on CPU]
> > >
> > >  We provide a detailed breakdown of DesignX’s training time per training step in the following table:
> > >
> > > | Agent-1 (7.5k steps) |  |  |  | Agent-2 (2.2M steps) |  |  |  |
> > > | --- | --- | --- | --- | --- | --- | --- | --- |
> > > | problem feature computation | workflow generation | BBO process | learning update | optimization prgress feature computation | parameter values inference | BBO process | learning update |
> > > | 2s | 0.95s | 20.01s | 0.03s | 0.001s | 0.02s | 0.04s | 0.09s |
> > >
> > > We would like to clarify that 6 days of training is not surprising since we involve a very large scale training problem set (9600 diverse problem instances). According to the training settings we provided in Appendix E.1, for Agent-1, it goes through 7.5k learning steps and require 2 days training. A large portion of per step training time is used for rollout the BBO processes given the workflow generated by Agent-1, since we have rollout 10 indepent runs to attain a subjective reward signal for this leaning step.  For Agent-2, the parameter control process allows immediate reward signal hence we use PPO train it with 2.2M steps. We train agent-2 3 times per 10 rollout steps, for per training step, we can still observe that BBO process is the most time-cunsuming part.
> > >
> > > To summarize, the true computational bottleneck of DesignX is not its deep learning framework, instead, is the inherent simulation cost in BBO optimization loop. This is also one of the major reason we train DesignX with CPUs, since BBO optimizers comprises many sequential logics that can not benefit from GPU’s matrix and parallel acceleration support. Besides, if we put agents on GPU and the BBO process in CPU, the CPU-GPU communication overhead is also too much. We hope this could clear your concern.
> > >
> > > ### [Discussion, 'design times' vs. the ‘actual computation overhead’]
> > >
> > > We understand the ‘counterintuitive feeling’ you have. As a machine learning, or more specifically, meta-learning task, MetaBBO researches such as our DesignX should be capable of leveraging GPU-like computational architecture to scale themselves towards wide range of applications. However, MetaBBO has its own characteristics, of which the most important one is that they include BBO optimization process within the bi-level methodology. Comparing with classic machine learning tasks and robotic RL tasks, BBO process is harder to couple with GPU-based scalability. This is due to the sequential nature of optimizers. We have to note that DesignX’s learning part can surely be deployed on GPU side. We are happy to see that recently, there are some initial exploration on matrix-based optimization [1] and jax-based parallism for optimization [2], which outlines DesignX’s future evolution path: by representing modules and algorithmic workflows with parallel-friendly blocks, the overall DesignX could be fully put on GPU to enjoy great scalability improvement.
> > >
> > > [1] Zhan Z H, Zhang J, Lin Y, et al. Matrix-based evolutionary computation[J]. IEEE Transactions on Emerging Topics in Computational Intelligence, 2021, 6(2): 315-328.
> > >
> > > [2] Huang B, Cheng R, Li Z, et al. EvoX: A distributed GPU-accelerated framework for scalable evolutionary computation[J]. IEEE Transactions on Evolutionary Computation, 2024.

---

> > > > ### Comment · Reviewer_kkQ7 · 2025-08-02
> > > > **Thanks the authors**
> > > >
> > > > Thanks the authors for their follow-up, which addresses most of the concerns.
> > > >
> > > > [Revise claims of the design time improvement]
> > > > Thank you. This would make the claim much fairer. I agree with the authors that an important future work could be comparative study with diverse human participants. This would be a good window-breaker to fully reveal the usefulness of designing automatic BBO algorithms. I suggest the authors to extend the setting mentioned in their observations and to establish a solid comparison in the future work.
> > > >
> > > > [GPT-2 configuration]
> > > > Thank you for disclosing the detailed architecture. Include the detailed configuration will address this concern.
> > > >
> > > > [Computing time]
> > > > Thank you for providing the break down sheets for training / computing time of DesignX. As discussed before, this will be insightful for readers to understand the bottleneck of developing large-scale automatic BBO framework. I totally agree with the authors that the current hardware infrastructure design is not always dedicated for algorithmic effort like DesignX, where many of the core computation have not always been fully optimized via parallelism. Adding this table as well as the related discussion would address this concern.  The recent work on discussing the potential computation optimization mentioned by authors looks interesting, please include them into the related works / discussion.
> > > >
> > > > Based on the discussion, if the authors agree to make these changes accordingly, I am happy to raise my score to 'Accept'. Is it allowed to provide a revision to the manuscript before discussion period ends? If so, I suggest the authors to do that.

---

> > > > > ### Author Response · Authors · 2025-08-03
> > > > > **Thanks for Reviewer kkQ7**
> > > > >
> > > > > Dear Reviewer kkQ7, we express our deepest gratitude for your consistently timely feedback. We kindly clarify that authors cannot submit revisions or supplemental material during the rebuttal and discussion period per conference policy. We promise to make the mentioned revisions immediately upon the revision window opening. We respectifully request you to check this update later and thanks for raising your score to 'Accept'.

---

### Decision · Program_Chairs · 2025-09-17

**Decision:**

Accept (poster)

**Comment:**

DesignX presents a dual-agent RL framework that searches a modular space of black-box optimiser components. One agent proposes an algorithmic structure, the other tunes hyperparameters, and the resulting designs are evaluated across a broad suite of synthetic and applied tasks. The idea is timely, the modular search space is well motivated, and the study is substantial. Reviewers generally found the contribution credible and useful, with performance that is competitive with strong human-crafted baselines such as CMA-ES in several settings. Overall, the paper can be accepted pending incorporation of discussion-phase commitments, as well as minor presentation fixes, including unambiguous diagrams of the two-agent architecture and clear documentation of baseline tuning.